# RNA-Seq of Phenotypically Distinct *Eimeria maxima* Strains Reveals Coordinated and Contrasting Maturation and Shared Sporogonic Biomarkers with *Eimeria acervulina*

**DOI:** 10.3390/pathogens13010002

**Published:** 2023-12-19

**Authors:** Matthew S. Tucker, Celia N. O’Brien, Alexis N. Johnson, Jitender P. Dubey, Benjamin M. Rosenthal, Mark C. Jenkins

**Affiliations:** 1Animal Parasitic Disease Laboratory, Beltsville Agricultural Research Center, Agricultural Research Service, United States Department of Agriculture, Beltsville, MD 20705, USAjitender.dubey@usda.gov (J.P.D.); benjamin.rosenthal@usda.gov (B.M.R.); mark.jenkins@usda.gov (M.C.J.); 2Department of State, Bureau of Consular Affairs, Washington, DC 20006, USA

**Keywords:** *Eimeria*, transcription, differential expression, coccidia, sporulation, development, oocyst, *Cyclospora*, control, vaccine

## Abstract

Strains of *Eimeria maxima*, an enteric parasite of poultry, vary in virulence. Here, we performed microscopy and RNA sequencing on oocysts of strains APU-1 (which exhibits more virulence) and APU-2. Although each underwent parallel development, APU-1 initially approached maturation more slowly. Each strain sporulated by hour 36; their gene expression diverged somewhat thereafter. Candidate biomarkers of viability included 58 genes contributing at least 1000 Transcripts Per Million throughout sporulation, such as cation-transporting ATPases and zinc finger domain-containing proteins. Many genes resemble constitutively expressed genes also important to *Eimeria acervulina.* Throughout sporulation, the expression of only a few genes differed between strains; these included cyclophilin A, EF-1α, and surface antigens (SAGs). Mature and immature oocysts uniquely differentially express certain genes, such as an X-Pro dipeptidyl-peptidase domain-containing protein in immature oocysts and a profilin in mature oocysts. The immature oocysts of each strain expressed more phosphoserine aminotransferase and the mature oocysts expressed more SAGs and microneme proteins. These data illuminate processes influencing sporulation in *Eimeria* and related genera, such as *Cyclospora*, and identify biological processes which may differentiate them. Drivers of development and senescence may provide tools to assess the viability of oocysts, which would greatly benefit the poultry industry and food safety applications.

## 1. Introduction

Coccidian parasites of livestock, including those caused by species of *Eimeria, Toxoplasma*, and *Cyclospora,* cannot induce infection immediately upon excretion by their prior host; instead, they undergo an extrinsic development (termed sporulation) necessary to become infectious [1,2]. Temperature and humidity regulate the pace of sporulation, which varies among species (and possibly among strains or genotypes of such parasites), thereby influencing the rapidity of the transmission cycle. Better understanding the common regulators of extrinsic development might provide better means to slow or stop extrinsic development, to the benefit of public health and livestock husbandry. If common developmental differences govern both extrinsic and intrinsic parasite reproduction, then differences evident during sporulation may induce, or parallel, phenotypic differences expressed as parasite strains establish and perpetuate infection in the host. This motivated our curiosity to study the sporulation differences among two strains that differ markedly in their virulence as enteric pathogens.

Like the enteric human pathogen *Cyclospora cayetanensis,* species of *Eimeria* cause enteric disease. A 2016 study estimated the global cost of poultry coccidiosis to exceed $14 billion (U.S.) globally each year [3]. Seven universally recognized species of *Eimeria* infect chickens and each parasitizes specific regions of the intestinal tract. The species *E. maxima*, *E. tenella*, and *E. acervulina* present the greatest disease risk and have the greatest economic importance [4]. Therefore, most *Eimeria* vaccines comprise these species [4,5,6]. *Eimeria maxima* parasitizes the jejunum of the small intestine [7] but can move to other locations during heavy infection. This parasite causes diarrhea and thickened intestinal walls, and it alters intestinal nutrient uptake [6,7]. *Eimeria maxima* can have insidious effects such as causing lower weight gain and poorer feed conversion efficiencies, but also can predispose chickens to necrotic enteritis (NE) caused by the colonization of the toxin-producing bacterium *Clostridium perfringens* [6,8].

*Eimeria* spp. infecting poultry induce specific host responses generally mediated by T cell responses. Little is known of intraspecific strain differences. *E. maxima* is highly antigenic; administering only a few oocysts can suffice to induce almost complete immune protection to homologous challenge [9]. Yet, this parasite also exhibits immunological variability [9,10,11,12,13,14,15,16,17,18]. Given the lack of cross-protection among most strains, vaccines must incorporate multiple strains for each species [4,11,19].

Two strains of *E. maxima* (APU-1 and APU-2) differ in pathogenesis in vivo [18,20]: the APU-1 strain is consistently more pathogenic than the APU-2 strain. Chickens infected with APU-1 gained 20–25% less weight than chickens infected with APU-2, when inoculated with as few as 1000 oocysts. The lesion scores were also 1–2 points higher in chickens infected with APU-1 compared to chickens infected with APU-2. APU-1 also appeared more fecund, producing more oocysts. This phenomenon was more pronounced at lower inoculation doses, as chickens infected with APU-1 produced ~2.5 times more oocysts than chickens infected with a similar dose of APU-2. A recent study [20] also showed that APU-1 invades host cells about twice as efficiently as APU-2.

The biological basis of these differences remains unclear. As in other species of *Eimeria*, asexual development (schizogony) precedes sexual development (gametogony). Recently, we reported 4–5 generations (multiplicative cycles) of schizogony before formation and the excretion of oocysts 4 days post-oral inoculation with *E. maxima* oocysts [7].

Here, we sought to understand whether and how these strains of *E. maxima* may differ in their extrinsic development, the period during which excreted oocysts sporulate to their infectious stage. We built upon omics approaches capable of identifying proteins, transcripts, and gene sequences that demarcate strains and the developmental stages of *Eimeria* [21,22,23,24,25,26,27,28]. Such genetic determinants may help explain phenotypic differences and describe life cycle development [25,28,29,30,31,32,33,34,35,36]. By delineating markers of viability during *E. acervulina* oocyst development shared with other species of *Eimeria* and *Cyclospora,* we aimed to understand what factors uniquely contribute to development in each species and strain. This will help identify developmental processes that govern coccidian development [37].

To this end, we employed comparative transcriptomics to delineate shared and distinct extrinsic development in two strains of *Eimeria* exhibiting contrasting virulence in their natural host. Understanding conserved developmental pathways may suggest general means to manage parasite harms (by, for example, retarding or preventing maturation to the infectious state); conserved features of development may also provide practical tools for assessing the viability of parasite stocks employed for vaccination. Understanding features of development unique to strains may illuminate the biological basis of such phenotypic variation, including differences important to animal health and productivity (such as virulence differences between the APU-1 and APU-2 strains of *E. maxima*). Therefore, we harnessed RNA sequencing (RNA-Seq) to understand their transcriptional differences during sporulation.

## 2. Materials and Methods

### 2.1. Ethics Statement

Animal experiments were performed following the protocol (22–06) approved by the BARC Institutional Animal Use and Care Committee, the United States Department of Agriculture. The chickens utilized in this study exhibited no outward signs of severe disease over the course of the study. After the study’s conclusion, all chickens were humanely euthanized; all efforts were made to minimize animal suffering.

### 2.2. Parasites

To obtain parasites for sporulation studies, we used methods described in [37] with some modifications. Briefly, two groups of eight male broiler chickens (Hubbard/Ross HR708, Longneckers Hatchery, Elizabethtown, PA, USA) were infected via oral gavage with either 3000 sporulated oocysts of strain APU-1 or 12,000 oocysts of strain APU-2. On day six post-inoculation, feces from the birds infected with a given strain were harvested and pooled. Unsporulated oocysts were isolated from chicken feces via flotation on a saturated salt solution and then washed 3–4 times with water. The final oocyst preparation for each strain was resuspended in 2% *v*/*v* potassium dichromate in a sterile 1 L flask. The approximate yield of 3 × 10^8^ oocysts was split equally into three flasks (denoted replicates A, B, and C) and incubated in a 29 °C shaking water bath aerated using an aquarium pump to promote sporulation.

An aliquot of 5 × 10^6^ oocysts was removed from each replicate flask at the beginning of the time course (T0) and then every six hours up to 24 h (T6, T12, T18, T24) and then every 12 h thereafter (T36, T48)). At each time point, the oocysts were centrifuged at 2400× *g* for 5 min at 4 °C and washed with deionized water to remove excess potassium dichromate. The oocysts were then incubated with sodium hypochlorite (6%) for 15 min with rocking agitation at room temperature to remove exogenous microbial contamination. After bleach treatment, samples were washed with deionized water to remove residual bleach. After washing, oocysts were resuspended in TRIzol (Thermo Fisher Scientific, Waltham, MA, USA) and frozen at −80 °C.

### 2.3. Microscopy

Aliquots of the oocysts at each time point were examined using a hemacytometer at 400× with an AxioScope microscope (ZEISS, Oberkochen, Germany). The percent sporulation was determined by counting the number of sporulated oocysts (those with four completely formed sporocysts) and unsporulated oocysts in multiple fields. We examined the extent of sporulation at additional time points (T30 and T42, but RNA from the oocysts was not isolated). To obtain optimal photomicrographs of developing oocysts, aliquots from each time point were transferred into a microcentrifuge tube and spun at 3400× *g* for 2 min. The oocyst pellets were washed once with water and final pellets were resuspended in 100–200 µL water before visualization. Images of representative oocysts at each time point were captured using a ZEISS Axioscope camera and the AxioVision imaging software version 4.9.1.0 (400× total magnification). In this study, we define parasite cohorts enriched for unsporulated oocysts as immature (T0) and cohorts enriched for sporulated oocysts as mature (T36).

### 2.4. Sample Preparation for RNA-Seq

Oocysts frozen in TRIzol were thawed on ice and 2 mL was transferred into a glass homogenizing tube. The suspension was ground using a Teflon pestle for 3–4 cycles of 25 grinds each. The samples were periodically examined using microscopy to monitor the progress of oocyst and sporocyst breakage. Homogenates were transferred into 2 mL RNAse-free microcentrifuge tubes, mixed with chloroform, and incubated at room temperature for 3 min. The tubes were centrifuged at 13,500× *g* for 15 min at 4 °C. The upper aqueous phase was transferred into new tubes and mixed with an equal volume of 70% ethanol. The total RNA was then isolated using a RNeasy Kit (QIAGEN, Germantown, MD, USA) following the manufacturer’s instructions. The total RNA was quantified using a Qubit 3.0 fluorometer and Qubit RNA High Sensitivity kit (Thermo Fisher Scientific) and the quality was assessed using a Bioanalyzer 2100 and RNA 6000 Nano kit (Agilent Technologies, Santa Clara, CA, USA). The RNA samples were frozen at −80 °C. The total RNA was processed using an Invitrogen Turbo DNA-free kit (Thermo Fisher Scientific). The total RNA of suitable quality for RNA-Seq had an RNA Integrity Number (RIN) ≥ 7 when assessed using a Bioanalyzer 2100.

### 2.5. cDNA Library Construction and RNA-Seq

The library construction and RNA-Seq utilized methods described by [37] with some exceptions. For each strain, whole coding transcriptome libraries were produced from each of three replicates of oocysts collected at each time point. Approximately 500 ng of total RNA was used as input using the Illumina Stranded mRNA Prep ligation kit (Illumina, San Diego, CA, USA) for cDNA library preparation. The cDNA libraries were quantified using Qubit, and the size distribution of libraries was characterized for each library using a Bioanalyzer 2100 and High Sensitivity DNA kit. The average library size for each sample was determined (ranged from 302 to 388 bp) and normalized for stoichiometric balance. The indexed libraries were then pooled and run on an Illumina NextSeq 2000 using a NextSeq 2000 P3 kit (300 cycles). After the run, reads were automatically converted into .fastq format using Bcl2Fastq (Illumina) and demultiplexed. The sequencing run resulted in four .fastq files per sample, containing two paired reads per lane. Forward and reverse reads were concatenated to produce single forward and reverse read files.

Raw, unpaired Illumina sequencing read files were used for analysis. Reads were imported into Geneious Prime 2023.04 (Biomatters Inc., Boston, MA, USA), https://www.geneious.com (accessed on 10 November 2023), paired, and trimmed using the package BBDuk (version 38.84) to remove Illumina adapter sequences. The adapter sequences were trimmed from the right end using default settings. Additional settings included “Trim Low Quality”, both ends with minimum quality = 20, “Trim adapters based on paired read overhangs”, minimal overlap = 20, and “Discard Short Reads”, minimum length = 30 bp. The trimmed reads were aligned with the most current genome sequence for the *E. maxima* Weybridge reference strain (annotated NCBI assembly EMW001, [38]) using the Geneious RNA mapper. The expression for each mapped sample was calculated.

We used Transcripts Per Million (TPM) as the main expression metric, calculated by Geneious as (CDS read count × mean read length × 10^6^) ÷ (CDS length × total transcript count). Data tables were exported from Geneious as .csv files and further analyzed in Microsoft Excel (Redmond, WA, USA) to calculate descriptive statistics of the raw reads and TPM for the sample replicates. Data analysis and visualization was aided by Daniel’s XL Toolbox add-in for Excel, version 7.3.4, by Daniel Kraus, Würzburg, Germany (www.xltoolbox.net (accessed on 10 November 2023)). Correlation (Pearson’s) of the normalized RNA-Seq mapped reads (log_2_ transformed) among strains and replicates was depicted as a heat map constructed in R version 4.3.1using the packages ggplot2 [39] and reshape2 [40]. Transcriptional bias (based on mean TPM from three replicates per time point) as previously described [37] was calculated at each time point for each strain.

### 2.6. Comparing Differential Gene Expression between Developmental Stages in Samples and Strain

The DESeq2 package [41], as implemented in Geneious Prime 2023.04, calculated the pairwise differential gene expression of time points in grouped replicates. This allowed us to compare the expression of all time points between strains as well as time points during sporulation (within and between strains). We examined differential expression using the log_2_ ratio of genes and adjusted *p*-values, which were used for downstream analyses. We used thresholds of >1.5 or <−1.5 log_2_ fold change (FC) with an adjusted *p*-value <0.05 to determine significantly differentially expressed genes (DEGs). Volcano plots depicting the log_2_-FC-vs. −log_10_-adjusted *p*-values (Adj. *p*-value) for DEGs were constructed in the R package Enhanced Volcano [42]. Correlation plots comparing log_2_ FC of DEGs between strains were constructed in the R package ggplot2.

### 2.7. Homology Searching and Functional Genomics Analysis

Functional information on genes of interest was derived from the annotations available for the *E. maxima* Weybridge reference strain in NCBI (assembly EMW001) and ToxoDB v62 ([43]). To elucidate the additional functions of genes, we utilized the functional analysis module in OmicsBox 3.0.27 (BioBam Bioinformatics, Valencia, Spain). Our Blast2GO^®^ workflow (for Phylum Apicomplexa or *Eimeria* spp.) was previously described in [37]. Other custom Blast2GO settings included an AnnotationE-value-Hit-Filter of 0.001, and all available Interpro Families, Domains, Sites, and Repeats. The rest of the workflow was run with default settings. Additionally, the eggNOG (evolutionary genealogy of genes: Non-supervised Orthologous Groups) [44] mapper (version 2.1.0 with eggNOG 5.0.2) inferred orthology relationships, gene evolutionary histories, and functional annotations to improve the Blast2GO sequence characterization. We used KEGG (Kyoto Encyclopedia of Genes and Genomes) [45] in pathway analysis settings, which enabled biochemical pathway enrichment analysis. KEGG was used with default settings and linked pathways via Enzyme Commission (EC) numbers. The Blast2GO results and lists of GO annotation terms and KEGG pathways were exported from OmicsBox and analyzed further in Microsoft Excel. Additional information for gene homologs and orthologs/paralogs and KEGG pathways (exact EC matches only) was found by searching the data sets in ToxoDB.

### 2.8. RT-qPCR

RNA-Seq estimates of gene expression were validated on a selection of genes using reverse transcriptase quantitative real-time PCR (RT-qPCR), following procedures described by [37]. Individual RT-qPCR reactions were prepared using SsoAdvanced Universal SYBR Green Supermix (Bio-Rad, Hercules, CA, USA), using 200–500 nM of each primer (Appendix A), and 1 µL of diluted cDNA to a total volume of 10 µL. Expression of *EMWEY_00042350* (Beta tubulin) was used as a reference to normalize expression levels of target genes. For determining the expression of constitutively expressed genes during sporulation, the abundance of mRNA in each replicate and measured time point was compared to that in the unsporulated oocyst samples (T0). Gene expression was estimated for the reference and target genes after averaging the Cq values for each replicate at each time point. The fold change in expression was calculated using an efficiency-corrected relative expression method [46]. The log_2_ FC of genes at each time point determined using DESeq2 (using mean of replicates for each RNA-Seq time point relative to T0) was compared to the log_2_ expression determined using RT-qPCR (mean of three trials, compared to T0). Each experiment was performed three times, and the mean of expression change was calculated for each gene and time point. All primers (Appendix A) were designed using NCBI Primer-BLAST [47] and synthesized by Integrated DNA Technologies (Coralville, IA, USA).

## 3. Results

### 3.1. Development of E. maxima Oocysts via Microscopy

We determined the percentage of oocyst sporulation of each *E. maxima* strain every six hours during the entire developmental time course (up to 48 h). From hours 6 and 12, the number of sporulated oocysts in APU-1 increased ~4.5 fold whereas it increased ~3-fold for APU-2 (Figure 1A). By 24 h, most oocysts of APU-2 destined for sporulation (~90%) had sporulated; APU-1 reached this benchmark six hours later. Each strain underwent similar development at each time point, but the proportion of sporulated oocysts (in which four distinct sporocysts were visible) was greater in APU-2 at hours 18 and 24. Each strain achieved maximum sporulation by hour 30. Figure 1B depicts the representative developmental stages during the sporulation course.

### 3.2. Transcription of E. maxima Oocysts during Sporulation Correlates Highly with Few Exceptions

Overall, a total of ~1.29 billion 100 bp paired-end reads (after trimming) were generated from the 42 samples (derived from three replicates at each time point, Appendix A). The number of reads per replicate and time point ranged from 15,969,320 to 44,080,524 (mean 29,402,440 ± 6,837,261) in APU-1 and 21,420,910 to 69,735,510 (mean 31,899,659 ± 12,823,456) in APU-2. The percentage of reads mapping to the *E. maxima* reference genome averaged ~95% (median = 95.5% for APU-1 and 95.3% for APU-2) in each strain (ranged 93.9–96.2% for APU-1 and 94.1–95.8% for APU-2).

We first analyzed the expression of 6057 annotated genes for each strain and replicate by examining the correlation of log_2_-transformed mapped reads (Appendix A). The Pearson’s correlation among replicates within each strain was high (*r* > 0.91, Appendix A). In each strain, the correlation between time points generally eroded with time, progressing from immature (T0) to mature oocysts (T36). For APU-1, gene expression correlated least when comparing T0 to T36 (*r* = 0.725–0.760). For APU-2, weaker temporal correlations were observed when comparing T0 to T18 through T48 (generally below *r* = 0.79, ranged *r* = 0.677–0.783).

The correlation of mean log_2_-transformed reads (from three replicates per time point, Appendix A) ranged from *r =* 0.749–0.979 in APU-1 and *r* = 0.730–0.984 in APU-2 (Figure 2). Within a strain, the next strongest correlation in gene expression generally occurred among temporally adjacent pairs (usually spanning 6–12 h). Between strains, the overall mean transcription of genes over time (log_2_ transformed reads per gene in Appendix A) correlated well (*r* = 0.981). Sometimes, a strain was more similar (or about the same) to itself at another time point than the exact time point of the other strain. For example, APU-1.T48 and APU-1.T36 had *r* = 0.957 but APU-1.T48 and APU-2.T48 had *r* = 0.93 (Figure 2). The discordance between strains (as measured by % reduction in correlation for any pair of time points) was greatest when comparing the expression at hour 48 to the expression at hour 0; at this pair of intervals, expression in APU-1 correlated 14.92% more than in APU-2. Comparing strains, the expression differed least when comparing hour 6 to hour 0 (0.02%). Thus, strain-specific differences in gene expression did not arise immediately.

Key asynchronous comparisons at hours 18, 24, and 48 (see Figure 2, boxed area) underscored differences in maturation rate, reinforced by the evidence from microscopy (Figure 1). Generally, when strains did not correlate most at the same time point, the best time point was normally ±6–12 h from this time. These exceptional correlations suggested the slower development of APU-1 than APU-2:APU-2′s expression at hour 18 best resembled APU-1′s at hour 24 (*r* = 0.976); by contrast, APU-1 at hour 18 best resembled APU-2 at hour 18.Expression in APU-1 at hour 48 correlated best with expression in APU-2 at hour 36 (*r* = 0.942).

Interestingly, the greatest correlation in expression between the strains when sporulated oocysts predominated (after T18) was at hour 36 (*r* = 0.973), and it ebbed slightly by hour 48 (*r* = 0.93). Therefore, evidence from microscopy and transcription established hour 36 as a stable interval for most oocysts to complete sporulation. In general, one strain’s expression of a given gene at a given time point generally predicts that gene’s expression in the other strain, but the quality of this prediction ebbs between 18 and 24 h, when development in APU-2 outpaces development in APU-1.

### 3.3. RNA-Seq in E. maxima Strains Enabled Analysis of Highly Expressed Genes and Transcriptional Bias

Previous data [37] identified a subset of highly expressed genes throughout *E. acervulina* sporulation that serve as markers of oocyst viability. Here, we sought evidence for such transcripts in *E. maxima* strains APU-1 and APU-2. We used Transcripts Per Million (TPM) as a normalization method, to summarize the expression of the 6057 annotated genes. Appendix A contains the mean and standard deviation (SD) of TPM of replicates at each time point.

Initially, we analyzed how uniformly genes contribute to the transcriptome during oocyst development in each strain. We surmised that many of the annotated genes in a strain would contribute relatively few transcripts (indeed, the mean TPM for 6057 genes = 165.10, or <0.017% of total transcripts). The number of genes expressed with >100 TPM at each time point ranged from 723 to 1332 (11.94 to 21.99% of the genes for APU-1) and 718 to 1381 (11.85 to 22.80%) for APU-1 (Table 1); therefore, ~5000/6057 (~83%) annotated genes contributed <100 TPM in each strain. This is in line with our previous report for *E. acervulina*, where <15% of genes during sporulation contributed >100 TPM [37].

To illustrate the range of transcription in strains during sporulation, the lowest expressed gene in APU-1 was at hour 36 (25,751 TPM), whereas the most transcribed gene was at hour 0 (64,395 TPM) (Table 1; this encompassed a range of 2.57–6.44% of total transcripts). For strain APU-2, expression of the most transcribed ranged from 26,560 TPM (at hour 24) to 48,731 (hour 18) (a range of 2.66–4.87%).

As previously established for transcription in *E. acervulina* [37], we conducted an analysis of transcriptional bias (excess contribution of abundant transcripts). Bias was examined for all genes and then only those with >100 TPM. At each time point (for each strain), transcription was slightly more biased in genes expressed with >100 TPM (Figure 3). The greatest transcriptional bias was observed at 18 h, later than its maximum in *E. acervulina* (at 4 h). The bias (whether examining all genes or only those with TPM > 100) did not change more than ~3.5-fold from its maximum at T18 to its minimum (at either T0 or T48). The amount of bias observed for all genes at each time point and those with >100 TPM correlated well (*r* > 0.999, data not shown) within each strain.

The degree of bias correlated, temporally, among strains (*r* = 0.88 comparing all genes; *r* = 0.874 for genes with >100 TPM). The greatest difference in bias occurred initially, appearing greater in APU-1 than in APU-2; from hours 0 to 6, transcriptional bias remained largely unchanged in APU-1 but increased ~2.5-fold in APU-2. Bias in APU-2 exceeded bias in APU-1 only at hour 6, perhaps due to the genes associated with its accelerated development during that interval.

### 3.4. Genes Expressed at High Levels throughout Sporulation Are Generally Similar between E. maxima Strains and E. acervulina

A major interest of ours is the identification of markers of viable and infectious coccidian oocysts. Therefore, we examined which transcriptional activities occurred throughout the entire sporulation time course. At any given time, as few as 74 genes contributed >1000 TPM (at hours 12, 18, Table 1). Previously, for *E. acervulina,* we established an expression threshold of 1000 TPM to identify amplified biomarkers, and here we applied this same criterion to *E. maxima* strains, identifying genes exhibiting constitutive expression of at least >1000 TPM throughout sporulation.

Of the 6057 annotated genes, 63 (10.4%) were constitutively expressed in APU-1 and 59 (9.74%) were constitutively expressed in APU-2; 58/6057 (9.58%) genes were constitutively expressed *in each strain*. In this group of 58 genes, 36.2% (21/58) are annotated only as hypothetical proteins in NCBI and ToxoDB. The mean TPM for each gene over time (Appendix A) was generally similar among strains; exceptions included genes (e.g., *EMWEY_00024920*, *EMWEY_00025290*, *EMWEY_00034320*, *EMWEY_00057080*) with >1.4-fold TPM differences between strains.

A general temporal trend held in the expression of these genes: the mean expression of the 58 genes peaked at 12–18 h in each strain and then decreased steadily (Figure 4). Large variation existed for some genes during sporulation due to high expression ranges at certain time points (Appendix A). However, over 50% of the genes had an overall mean TPM < 10,000 with relatively little variation (~<4000 SD for mean TPM). The top 10 genes (by mean TPM over time) for each strain were expressed with at least 17,000 TPM. Nine of these genes were common to each strain (e.g., *EMWEY_00006260*, *EMWEY_00042090*) and they contributed more to the total number of transcripts and underwent greater change over time.

To look more broadly, we compared these constitutively expressed genes in *E. maxima* to our previously reported *E. acervulina* RNA-Seq data [37]. Of note, most such genes (55 of 58) had a high BLASTP hit in *E. acervulina* (three exceptions are bolded in Appendix A), and of the 55 that could be matched, 41 (~75%) were also constitutively expressed in *E. acervulina*. Of the 58 genes expressed >1000 TPM in *E. maxima*, 14 (24.1%) had homologs in *E. acervulina* that did not meet equivalent expression criteria (bolded, Appendix A). Therefore, 17 of the 58 constitutively expressed genes in the *E. maxima* strains either could not be matched using BLAST or had a match in *E. acervulina* that was not deemed constitutively expressed in that species. Interestingly, the mean expression of the 41 matched constitutively expressed genes we identified from *E. acervulina* had maximum mean TPM at four hours (opposed to 12–18 h in *E. maxima*, Figure 4). These data show that most of the *E. maxima*-identified genes had homologs previously identified as constitutively expressed in *E. acervulina,* defining conserved genes important throughout sporulation.

#### Understanding the Functionality of Constitutively Expressed Viability Markers

To further characterize promising *E. maxima* candidate biomarkers for viability, we focused on the genes shared between APU-1 and APU-2 that were expressed with at least 20,000 mean TPM over the entire time course (eight genes, italicized in Appendix A). Two of these genes encode cation-transporting ATPases (*EMWEY_00006260* and *EMWEY_00006270*). Their homologs in *E. acervulina*, *EAH_00004110* and *EMWEY_00004100*, were also constitutively expressed genes we previously identified with high expression (>26,000 TPM; Appendix A).

In this group of eight genes, other functionally annotated genes encoded a zinc finger DHHC domain-containing protein (*EMWEY_00000500*). Its homolog in *E. acervulina* (*EAH_00034270*) was also a very highly expressed constitutive gene. The remaining highly expressed and shared *E. maxima* genes (*EMWEY_00003320*, *EMWEY_00033410*, *EMWEY_00042090*, *EMWEY_00057080*) encode hypothetical proteins. Of note, the homolog of *EMWEY_00042090* in *E. acervulina* (*EAH_00037050*) had the second highest overall mean TPM (>30,000, Appendix A). We could not identify a homolog for *EMWEY_00057080* (the highest overall expressed gene in APU-1) in *E. acervulina* (Appendix A). As we found previously for *E. acervulina*, many constitutively but modestly expressed genes in *E. maxima* have housekeeping functions.

In the *E. maxima* NCBI reference strain, 3926/6057 (64.8%) genes are annotated only as hypothetical proteins (as is true for other *Eimeria* spp.). Compared to our prior analysis of expression in *E. acervulina*, comparatively few hypothetical proteins occur among the constitutively expressed *E. maxima* genes (21/58 [36.2%] vs. 26/53 [49.1%] in *E. acervulina*). To facilitate an improved functional annotation of the hypothetical proteins, we used Blast2GO to search gene homologs Apicomplexa-wide and find related proteins. This search succeeded in identifying a more informative protein hit for 10 of the 21 genes initially identified as encoding hypothetical proteins (bolded in Appendix A). This provided more informative annotation (membrane or transmembrane proteins) for three of the eight highly expressed constitutive genes in the *E. maxima* strains (bolded and italicized, Appendix A).

Of the 58 constitutively expressed genes shared by the *E. maxima* strains, 50 (86.2%) were successfully annotated with at least one Gene Ontology (GO) term (Appendix A). With one exception (*EMWEY_00029830*, encoding a chemotaxis protein), the genes without GO terms encode hypothetical proteins. In total, there were 372 annotated GO terms (Appendix A), categorized into the subontologies of Biological Process (BP, 152), Molecular Function (MF, 119), and Cellular Component (CC, 101). Some genes with expression on the lower end of the spectrum (e.g., *EMWEY_00024250* [mean TPM > 5000 in each strain], *EMWEY_00045300* [mean TPM ~3000 in each strain]) had comparatively more GO terms.

The most frequent BP terms included those related to proteolysis, ubiquitin-dependent catabolic processes, protein modification, translation, transmembrane transport, and many encompassing actin activity. In addition, many terms pertained to regulation and cellular responses. The dominant MF terms pertained to binding (ATP, metal ion, DNA, actin, magnesium, phospholipid, etc.), ligase activity, ATP hydrolysis, endopeptidase activity, cytoskeleton, ribosome functions, and protein-cysteine S-palmitoyltransferase activity. Other terms pertained to a host of cellular functions. The most numerous CC terms included membrane, nucleus, cytosol, cytoplasm, actin, plasma membrane, nucleoplasm, assorted organelle terms, ubiquitinligase complex, and acetylcholine-gated channel complex. In addition, other terms were identified that can be grouped into ribosome structure and cytoskeleton/motility. A few genes encoding hypothetical proteins (that may have matched or not matched to a better annotation in other species) had associated GO terms. In general, many of the enriched terms were like those we identified for the constitutively expressed genes in *E. acervulina*.

Of the higher expressed genes shared by both strains described above, *EMWEY_00006260* and *EMWEY_0006270* had GO terms pertaining to membrane and mitochondrion (Appendix A). *EMWEY_00000500* had eight annotated GO terms including: protein targeting, peptidyl-amino acid modification, protein palmitoylation, protein-cysteine S-palmitoyltransferase activity vacuole, endoplasmic reticulum, Golgi apparatus, and plasma membrane. *EMWEY_00033410* (hypothetical protein) had homology to a membrane protein of *Besnoitia besnoiti* (Appendix A) and GO terms of tRNA aminoacylation for protein translation, nucleotide binding, aminoacyl-tRNA ligase activity, and membrane. *EMWEY_00042090* also had similar BLAST homology to a membrane protein of other Apicomplexans and the GO term membrane. Two genes encoding hypothetical proteins (*EMWEY_00003320* and *EMWEY_00057080*) did not have more informative hits in Apicomplexa species but *EMWEY_00057080* was associated with the GO terms histone methylation and metal ion binding.

Of the 58 constitutively expressed genes, 22 were involved in at least one enzymatic pathway using Blast2GO (Enzyme codes [ECs], Appendix A). Many genes below <10,000 overall mean TPM were among the lowest expressed over the time course, but some eclipsed >20,000 mean TPM. We also applied KEGG pathway analysis to identify large metabolic pathways for all genes. Overall, 12 genes had associated KEGG pathways (Appendix A); therefore, not all genes with ECs could be linked to a KEGG pathway. We also noted that some of these genes were among those with a larger number of GO annotations than others, as described above. The major pathways defining these genes included thiamine metabolism, purine metabolism, and glycerophospholipid metabolism. *EMWEY_00051720* (mean TPM ~6000 in both strains) was linked to an assortment of pathways (found from ToxoDB). Eleven of twelve genes contributing to KEGG pathways matched to homologs in *E. acervulina* that we previously found were also constitutively expressed by the same criteria (the exception was *EMWEY_00024920* = *EAH_00006770*). These 11 genes essentially were linked to the same KEGG pathways (Appendix A).

The highest expressed gene linked to a KEGG pathway differed in each strain. For APU-1, it was *EMWEY_00025290* (mean TPM = 25,192, linked to terpenoid backbone biosynthesis). For APU-2, it was *EMWEY_00011310* (mean TPM = 17,340, nicotinate and nicotinamide metabolism, alanine, aspartate, and glutamate metabolism; taurine and hypotaurine metabolism). The homolog to *EMWEY_00011310* in *E. acervulina* (*EAH_00019500*) was also a highly expressed constitutive gene and had the same KEGG pathways (Appendix A). Only one of the genes annotated as encoding a hypothetical protein (*EMWEY_00024920*) had an associated KEGG pathway (PD-L1 expression and PD-1 checkpoint pathway in cancer).

These data reinforce the idea that modestly and constitutively expressed genes encode enzymes in metabolic pathways that underlie basic housekeeping functions. There is a conservation of genes and similarity between species of *Eimeria*, and possibly among coccidia more generally. While we could identify possible homologs of hypothetical proteins (some highly expressed in both strains) in other Apicomplexan species, little more could be gleaned about their function.

### 3.5. Few Genes Contributed Many Transcripts throughout Sporulation to a Significantly Different Extent in APU-1 and APU-2

We sought biomarkers that might differentiate the two *E. maxima* strains by identifying significantly DEGs throughout sporulation. We searched for genes undergoing >1.5 or <−1.5 log_2_ fold-change (FC) with an adjusted *p* < 0.05 (Figure 5). Of 204 such genes identified, most contributed minimally (<100 TPM) to the total transcript pool. Ten genes differing in expression among strains contributed, on average, >100 TPM in each strain (Table 2). Genes particular to APU-1 were all upregulated in comparison to APU-2; eight genes were upregulated in APU-2. Two genes (*EMWEY_00007700* and *EMWEY_00023530,* italicized in Table 2 and Appendix A) surpassed 100 mean TPM in each strain (and exceeded 1000 TPM at some time points in APU-1) but underwent >1.5 log_2_ FC in expression when comparing strains. These two genes generally followed the same temporal transcriptional pattern. Other genes particular to each strain were generally expressed with <500 TPM during sporulation, although they were sometimes >6-fold (log_2_) differentially expressed between strains.

As a proportion of the total transcripts, certain genes contributed more than 4-fold more (on average) in APU-1 than APU-2 and were upregulated >2-fold (log_2_) (*EMWEY_00023530*, *EMWEY_00041620, EMWEY_00048910*, *EMWEY_00056530*) (Table 2). Conversely, *EMWEY_0005470* (encodes a Surface Antigen [SAG] family member gene) had ~3 times higher log_2_ FC in APU-2 than in APU-1 (and exceeded a mean TPM of 1000); the strain-to-strain TPM difference was greatest during intervals of active sporulation. Genes *EMWEY_00008090* and *EMWEY_00058460* were both upregulated >6-fold (log_2_) in APU-2 with very high differences in the overall mean TPM (expression of these genes was <10 TPM at all time points in APU-1). The genes in APU-2 sometimes achieved >200-fold TPM difference compared to APU-1 at T36. They may be useful in the identification of strains and identifying oocysts at stages during sporulation (see below).

For APU-2, 4/10 of the genes identified encode hypothetical proteins. Blast2GO did not identify a more informative match for these genes (Appendix A). In total, nine genes were associated with 61 GO terms and the majority associated with genes *EMWEY_00023530* (see above)*, EMWEY_00036080* (thioredoxin), *EMWEY_00039050* (Ovarian Tumor Unit [OTU]-like cysteine protease domain-containing protein), and *EMWEY_00058460* (elongation factor 1-alpha, [EF-1α]).

The GO terms for genes unique to APU-2 (Appendix A) can be summarized as follows. *EMWEY_00036080* has terms pertaining to endoplasmic reticulum and Golgi apparatus structure, protein targeting, transport, and folding. *EMWEY_00039050*’s GO terms were related to protein deubiquination and peptidase activity. *EMWEY_00058460* was associated with the most GO terms and these were of multiple types including tRNA transport, GMP and GTPase binding, ATP activity, actin assembly, and translation processes. Although this gene (and a few others) had EC numbers identified using Blast2GO, only *EMWEY_00058460* had linked KEGG pathways in our analysis (Appendix A). These pathways were related to drug metabolism, antibiotic synthesis, and metabolism of purines, sulfur, selenocompounds, and thiamine. These pathways were similar in the *E. acervulina* homolog for this gene (*EAH_00032440*), which had a very low mean overall expression throughout sporulation. The highest expressed gene (*EMWEY_00005470*) had a single GO term of “membrane” but no other useful information was identified. All the 10 genes had homologs we could identify in *E. acervulina*.

Of the 10 DEGs in APU-1, 9 encode hypothetical proteins. After Blast2GO analysis, only one of these genes (*EMWEY_00001070*) matched to a more informative annotated gene in another Apicomplexan species (SF-assemblin in *C. cayetanensis*, Appendix A). In total, 6 genes were associated with 23 GO terms with the majority (those with other than CC term of “membrane”) associated with genes *EMWEY_00001070*, *EMWEY_00012880*, and *EMWEY_00023530* (18kDa cyclophilin). GO terms for these genes generally fell into broad categories. *EMWEY_00012880* was associated with terms pertaining to cilia assembly and transport/targeting, and *EMWEY_00001070* was associated with cytoskeleton and cytoplasm. *EMWEY_00023530*, the highest overall expressed gene in the group, was associated with terms pertaining to cyclosporin A binding and protein folding and protein isomerization. Only this gene was associated with an EC number, but we did not identify a linked KEGG pathway.

Generally, genes expressed to differing extents in the strains of E. maxima contributed more to the total transcript pool than homologs of these genes in E. acervulina. Eight of the 10 APU-1 genes had homologs that we could identify in *E. acervulina.* These genes had relatively lower mean overall TPM (most <50 TPM) in *E. acervulina*. The homolog to *EMWEY_00023530* (*EAH_00035670*) was the highest expressed, eclipsing 160 TPM (Appendix A). The homolog to *EMWEY_00059740 (EAH_00003690*) was the highest expressed in *E. acervulina*. Interestingly, *EMWEY_00005470* (highest TPM in APU-2) had <1.0 mean TPM in *E. acervulina*. These genes encode SAGs, possibly indicating their increased importance in APU-2.

Overall, during sporulation, certain genes are expressed more in one strain than in the other. However, little additional information exists about the function of these DEGs: few such genes belonged to identifiable metabolic pathways; some were more prominently associated with certain GO terms. Interestingly, those genes identified by GO terms and KEGG pathways tended to be expressed at very low levels throughout sporulation. Therefore, we performed further analysis at specific time points in hopes of identifying more biomarkers important to each strain.

### 3.6. Differential Expression at Hours 36 and 48 during Oocyst Sporulation

Differences in the expression of genes at certain time points may drive, or reflect, developmental differences between strains. For example, at hour 36, genes *EMWEY_00023520* and *EMWEY_00056530* in APU-1 and *EMWEY_00005470, EMWEY_00008090*, and *EMWEY_00058460* in APU-2 were expressed much more in one strain than the other (sometimes up to 230-fold higher TPM, Table 2). Accordingly, we sought to identify genes especially responsible for the degree of correlation evident between strains at particular times (Figure 2). Against a general pattern of concurrent changes in expression, comparatively accelerated development in APU-2 was implied by cases where its expression profile correlated best with the expression of APU-1 6–12 h later (Figure 2). The number of significantly DEGs tended to increase as the transcription correlation between time points decreased (Table 3). This decrease in correlation also coincided with increasing DEGs at *concurrent* time points during sporulation between strains. Yet, the greatest number of DEGs occurred when comparing APU-1 at hour 18 with APU-2 at hour 24.

#### 3.6.1. DEGs at Hour 36 Demarcate Strain-Specific Biomarkers and Help Explain Expression Differences around Periods of Maximum Sporulation

We took special interest in genes whose expression drove a strong correlation between the strains at hour 36, by which time many oocysts completed the formation of visible sporocysts; we also sought to determine the causes of the evident ebbing of correlation by hour 48. Using the previously described criteria, we found 284 DEGs at hour 36 and 850 at hour 48 (Table 3). At hour 36, only three genes contributed >1000 mean TPM in APU-1 (*EMWEY_00006520, EMWEY_00007700, EMWEY_00023530*) and only one for APU-2 (*EMWEY_00005470*).

Considering any such genes contributing >100 mean TPM identified 21 genes in each strain, 10 were shared by both strains (italicized in Table 4). For APU-1, most of the DEGs identified (above) based on the average overall expression were also differentially expressed at hour 36 (except for *EMWEY_00001070* and *EMWEY_00057350*). For APU-2, only *EMWEY_00036080*, *EMWEY_00037810*, and *EMWEY_00042800* were differentially expressed overall but not at hour 36. Therefore, most genes differently expressed throughout sporulation were also differentially expressed at hour 36.

As determined for mean expression throughout sporulation, at hour 36, genes such as *EMWEY_00023530* and *EMWEY_00056530* were upregulated (>2-fold log_2_) in APU-1. All but six DEGs in APU-1 with >100 TPM encode hypothetical proteins. Generally, the genes with the highest TPM in APU-1 did not have the greatest differential expression vs. APU-2; however, genes such as *EMWEY_00005570* and *EMWEY_00023520* were upregulated ~3-fold (log_2_) compared to APU-2 (Table 4).

Except for one gene (*EMWEY_00014910*, bolded Appendix A), our Blast2GO pipeline did not identify any more informative annotations for APU-1 DEGs at hour 36. Also, most GO terms were associated with already annotated genes. Still, some upregulated genes had more than one term. For example, *EMWEY_00012880* (~2-fold log_2_ upregulated in APU-1, encoding a hypothetical protein) and *EMWEY_00023530* were associated with at least seven GO terms (see above). *EMWEY_00005570* (protein binding) and *EMWEY_00023520* (membrane) were each associated with a single GO term. *EMWEY_00006520* was associated with nine GO terms, including those relating to development and DNA binding and DNA damage repair. Three genes (*EMWEY_00006520*, *EMWEY_00017660*, *EMWEY_00052370*) had associated KEGG pathways according to our analysis; these were aminobenzoate degradation, amino sugar and nucleotide sugar metabolism, and porphyrin and chlorophyll metabolism, respectively (Appendix A).

At hour 36, 19 of 21 genes expressed to a significantly greater extent in APU-1 than in APU-2 had identifiable homologs in *E. acervulina*. As we found in the analysis of mean expression through time, these genes generally contributed a higher proportion of the total transcripts in *E. maxima* than their homologs in *E. acervulina*. However, the homologs of *EMWEY_00031340* and *EMWEY_00056980* (hypothetical proteins contributing <300 TPM at hour 36, downregulated vs. APU-2) were expressed to a much greater extent in mature oocysts of *E. acervulina* (italicized and red, Appendix A). In *E. acervulina,* they were DEGs between mature and immature oocysts [37].

As noted in strain APU-1, certain genes overexpressed throughout sporulation in APU-2 were also upregulated at hour 36 (compared to APU-1): for example, *EMWEY_00005470* and *EMWEY_00058460* (Table 4). Other genes were noticeable for a >2-fold (log_2_) upregulated expression change, including *EMWEY_00000860, EMWEY_00039050, EMWEY_00052050*, and *EMWEY_00059740* (note that *EMWEY_00000860* and *EMWEY_00052050* were genes not identified in our overall analysis).

Of the genes expressed more in APU-2 than APU-1 at hour 36, 10 of 21 encode hypothetical proteins. Except for one gene (*EMWEY_00007270*, bolded Appendix A), our Blast2GO pipeline identified no more informative annotated genes. Approximately half of the genes were associated with most of the GO terms. Some genes with related GO terms appeared similar—e.g., *EMWEY_00017480* and *EMWEY_00056980*—yet experienced only modest excess expression in APU-2. A group of genes (*EMWEY_00000860, EMWEY_00039050, EMWEY_00052300, EMWEY_00054680, EMWEY_00059760*) were associated with amino acid metabolism and protein synthesis/regulation GO terms. *EMWEY_00058460* was the highest upregulated DEG vs. APU-1 with 6.7-fold log_2_ differential expression.

The two SAG protein-encoding genes (*EMWEY_00005470*, *EMWEY_00059740*) identified in the overall analysis were also upregulated at T36. These genes have the highest expression during the time course (in each strain) at hour 36. It appears that mature oocysts in APU-2 may be characterized more by protein processes and SAG proteins (upregulated in APU-2, although they contributed >100 TPM in both strains). KEGG pathways were identified for five genes (Appendix A: *EMWEY_00006520* [see above], *EMWEY_00011930, EMWEY_00054680, EMWEY_00058460* [see above], *EMWEY_00059760*). These pathways concern thiamine/purine metabolism and amino acid and antibiotic synthesis, in addition to other pathways. These genes were all expressed <~500 TPM.

We found that 19 of the APU-2 genes also had homologs in *E. acervulina*. Two of the genes in common with APU-1 and discussed above had homologs in *E. acervulina* that were upregulated in mature oocysts, and *EMWEY_00011930* (heat shock protein homologous to *EAH_00002610*) was an additional one (italicized and red, Appendix A). The homolog was also expressed with >1000 TPM in *E. acervulina* and significantly differentially expressed vs. immature oocysts in that species.

#### 3.6.2. DEGs at Hour 48 Characterize the Divergence of Oocysts Compared to Hour 36

As described above, transcriptional profiles agreed best at hour 36 and eroded thereafter. Indeed, concurrent correlation appeared weakest at hour 48. (Figure 2). The number of DEGs, comparing the two strains, triples from hour 36 to hour 48 (from 284 to 850; Table 3). We sought to further understand the changes demarcating this departure (acknowledging that expression in one strain still explains 93% of the variation in expression in the other).

Of the 284 DEGs between strains at hour 36, 184 (64.8%) remained differentially expressed at hour 48. The log_2_ differential expression ratio for the remaining 100 genes ranged from ±2.7 to ±2.01 (APU-1 vs. APU-2), a tighter range than described in Table 4 (Appendix A). Genes with the largest magnitude in differential expression among strains tended to contribute fewer TPM. Only 12 of these 100 DEGs were upregulated in APU-1. Fourof the 100 genes contributed >100 TPM in APU-1, and only 6 contributed >100 TPM in APU-2 (these were described above and/or in Table 4). These four genes were also DEGs between strains at T36, possibly indicating their greater importance to that time point.

Of these 100 genes, 57 were annotated as hypothetical proteins and Blast2GO found 26/57 (~46%) had a more informative Apicomplexa BLAST hit (Appendix A). Among the 100 genes, some common annotations included DEAD-box domain-containing helicases, subtilases, translation initiation factors, and DNA damage proteins. The groups generally had similar differential expression profiles. Overall, 624 GO terms were found for the 100 genes. Many could be attributed to genes such as *EMWEY_00020830* (encoding thioredoxin reductase with 50 terms), *EMWEY_00043800* (hypothetical protein with 60 terms mainly pertaining to cellular transport and regulation), *EMWEY_00047940* (encoding a helicase protein, 20 terms), *EMWEY_00049690* (encoding an anti-silencing protein, 23 terms), and *EMWEY_00058910* (encoding a protein transport protein, 22 terms). These genes had a low mean TPM at hour 36 but comprised a higher proportion of total transcripts in APU-2 than in APU-1.

KEGG pathways were identified for 29/100 genes, most of which exhibited relatively low expression. *EMWEY_00006520* (one of the highest expressed genes in both strains, but upregulated in APU-1) was described above for the T36 DEGs. Most of the 100 genes had identifiable homologs in *E. acervulina*, but the expression between species varied (Appendix A). For example, *EAH_00017220* (homolog of *EMWEY_00038960*) was a highly constitutively expressed gene but its *E. maxima* homolog was expressed to a lesser extent. *EAH_00026150* was higher expressed in mature *E. acervulina* oocysts and its homolog in *E. maxima* (*EMWEY_00056980*) was on the higher end of expression at T36 in both strains.

Interestingly, most of the DEGs at hour 48 were not differentially expressed at hour 36 (666/850; 78.4%; Appendix A). Therefore, these genes appear to demarcate strain divergence after hour 36. The log_2_ FC of the 666 genes ranged from ±3.79 to ±3.71, an increase from the hour 36 DEGs. A greater relative number of genes were expressed >100 TPM in strains at T48 (92 in APU-1, 132 in APU-2) than at T36 and more genes eclipsed 1000 TPM. About 66% (438/666) of the genes were annotated to encode hypothetical proteins. Blast2GO analysis found better matches with more informative annotations for 166 of the 438 (~38%) genes originally encoding hypothetical proteins. As with the T36 genes, T48 had many genes that could be grouped into categories (e.g., ribosomal proteins, kinesins, ATPases, DNA polymerases, invasion proteins, subtilases/subtilisin, zinc finger proteins, etc.). Some were of similar nature to the T36 genes. However, genes encoding adaptin, chromosome segregation proteins, SAGs, myosins, DNA polymerases, mitochondria, and microneme proteins seemed unique to T48 oocysts. Again, the directionality of expression was usually similar for groups. Overall, there were 2904 GO terms identified for the 666 genes.

Although most of the T48 DEGs (not also differentially expressed in T36) were expressed to only a modest extent, 10 DEGs (4 upregulated, 6 downregulated in APU-1) and 23 genes in APU-2 (all upregulated vs. APU-1) had a mean TPM > 1000 at T48 (Appendix A). Genes with higher differential expression in APU-1 generally did not have as high TPM, however. The 10 APU-1 genes were significantly differentially expressed at least 1.6 log_2_ FC compared to APU-2; eight genes encoded hypothetical proteins. Genes *EMWEY_00025300* (hypothetical protein), *EMWEY_00036630* (hypothetical protein but had BLAST similarity to a protein kinase domain-containing protein in *E. mitis*), *EMWEY_00056350* (subtilase family serine protease), and *EMWEY_00070020* (ATP synthase beta chain) eclipsed 1000 TPM at T48 with at least >1.7-fold log_2_ higher differential expression than APU-2. GO terms were associated with *EMWEY_00036630* (such as protein kinase, ATP binding), *EMWEY_00056350* (including proteolysis, serine-type endopeptidase activity), and *EMWEY_00070020* (pertaining to translation, ribosomes). The *E. acervulina* homolog (*EAH_00048170*) of *EMWEY_00056350* was highly expressed throughout sporulation. The highest DEG was *EMWEY_00056940* (3.7-fold higher log_2_ FC in APU-1) with ~120 TPM in APU-1.T48 vs. ~2 TPM in APU-2.T48. This gene, originally annotated to encode a hypothetical protein, had BLAST similarity to a putative adenosine monophosphate deaminase in *Neospora caninum*. No KEGG pathways were identified for the 10 genes.

For the 23 APU-2 DEGs of interest, 14 exceeded >2000 TPM at T48. These genes were all upregulated vs. APU-1 with at least ~1.6 FC (log_2_). These included genes such as *EMWEY_00005600* (hypothetical protein), *EMWEY_00009150* (hypothetical protein, but BLAST similarity to *T. gondii* family D protein), *EMWEY_00016000* (hypothetical protein but BLAST similarity to an uncharacterized serine-rich protein C215.13-like in *C. cayetanensis*), *EMWEY_00016120* (hypothetical protein), *EMWEY_00016540* (hypothetical protein), *EMWEY_00017740* (hypothetical protein), *EMWEY_00028320* (hypothetical protein, but BLAST similarity to a putative autoantigen, coiled-coil vesicle tethering subfamily A protein in *T. gondii*), and *EMWEY_00030150* (hypothetical protein). Genes *EMWEY_00009150* and *EMWEY_00016000* were very highly expressed (>18,000 TPM) at T48. Several of these genes (and others with >2000 TPM) had multiple GO terms, including *EMWEY_00016120* (protein phosphorylation, kinase activity, ATP binding), *EMWEY_00017740* (transcription, nucleus, transcription regulation), *EMWEY_00023060* (encoding a serine protease inhibitor with terms pertaining to endopeptidases, regulation of endopeptidase activity), and *EMWEY_00031210* (encoding Hsp20 with terms pertaining to stress response, protein folding). Genes *EMWEY_00005600* and *EMWEY_00008410* (encoding an aldo/keto reductase family oxidoreductase) were the only genes with >2000 TPM at T48 with associated KEGG pathways: propanoate, inositol phosphate, amino acid metabolism, and sugar and other molecule catabolism, respectively. Generally, the highest DEGs in APU-2 had low expression but *EMWEY_00030150* (hypothetical protein) was ~3-fold log_2_ more highly expressed vs. APU-1 (and upregulated) with ~3700 TPM (no GO terms or KEGG pathways identified, however). We could identify *E. acervulina* homologs for almost all the genes with >2000 TPM but they were mostly on the low spectrum of expression. However, *EAH_00010650* (Hsp20, homolog of *EMWEY_00031210*) did reach >1000 TPM at T24 and was significantly differentially expressed vs. immature oocysts. Again, finding a clear association for genes underlying the time point expression between species seems challenging.

### 3.7. RNA-Seq Revealed Markers That Demarcate Strain-Specific Immature and Mature E. maxima Oocysts

We wished to find markers of immature and mature *E. maxima* oocysts, as we previously reported in *E. acervulina*. When comparing the overall differential expression (the log_2_ ratio between strains without cutoffs) of time points T36 andT0 between strains, there was a high correlation of DEGs (*r* = 0.90). Most genes underwent parallel expression changes in each strain: 2317 were upregulated and 2522 downregulated in each (upper-right or lower-left quadrants of Figure 6). Far fewer transcripts fall far from the origin in the upper-left (*n* = 740) or lower-right (*n* = 403) quadrants in Figure 6 (undergoing contrasting directions in temporal expression). Concurrent changes generally were of similar magnitude, but differences in the extent of temporal variation may be related to developmental differences between strains.

Generally low baseline expression prevented some genes, exhibiting large fold-changes in expression, from demonstrating statistical significance. Exceptions may signify strain-specific markers of immature or mature oocysts. Table 1 shows 2440 significantly DEGs between T36 and T0 in APU-1 and 2499 DEGs in APU-2 using our thresholds. Of these, 1871 DEGs were shared between strains. The log_2_ FC of these 1871 DEGs correlated highly (*r* = 0.96), reinforcing the conclusion that genes in strains undergo similar directional changes in their temporal expression.

We further examined genes undergoing differential expression in each strain at hours 0 and 36, starting with genes *not shared* by the pair of strains (569 DEGs for APU-1; 628 for APU-2). Of these, strong expression (>1000 TPM) at either hour 0 or 36 characterized 23 genes in APU-1 and four in APU-2 at hour 0 (Appendix A). At hour 36, strong expression characterized 19 genes in APU-1 and 2 in APU-2 (Appendix A).

Among the 23 high expression DEGs only in APU-1, most (16 = 69.6%) were downregulated at T0 (ranged −2.4–4.1 log_2_ FC). However, many of these genes contributed to high TPM at both hours 0 and 36 and were constitutively expressed in both strains (Appendix A), as well as in *E. acervulina*. One of the 23 genes (*EMWEY_00053520*, hypothetical protein) had a homolog in *E. acervulina* (*EAH_00034940*) that we previously identified as an upregulated immature oocyst biomarker (~2-fold log_2_ higher expressed in its immature oocysts). The seven remaining APU-1 DEGs were upregulated and not constitutively expressed. One gene may serve as an especially suitable marker of immature oocysts of APU-1 because its initial expression level was >4 log_2_ fold higher than its expression at hour 36 (*EMWEY_00052020,* an X-Pro dipeptidyl-peptidase domain-containing protein). Conversely, three genes specific to APU-2 underwent upregulation from hour 0 to hour 36, whereas one underwent downregulation. Their expression at T36 ranged from 1000 to 3000 TPM (lower than the APU-1 genes) and their log_2_ FC ranged from −2.3 to 2.13. All the genes were annotated but were not constitutively expressed. None were like the genes previously identified in immature oocysts of *E. acervulina*. These genes may identify immature oocysts of APU-2.

The same analysis for genes with >1000 TPM at T36 (DEGs vs. T0) in each strain identified 18 genes in APU-1 and 2 in APU-2 (Appendix A). For APU-1, almost all were the same as the shared constitutively expressed genes (plus *EMWEY_00006520* and *EMWEY_00036760*, Appendix A), owing to their high expression at both T0 and T36. Notably, *EMWEY_00006520* was expressed to an even greater extent in APU-1 than in APU-2 (log_2_ FC = 3.7). It was a DEG we identified between strains at T36 (Appendix A) and appeared unique to that time point (not differentially expressed at 48 h, Appendix A). Its higher expression vs. APU-2 at T36 (and T0 in APU-1) makes it an especially promising marker for mature APU-1 oocysts. Its homolog (*EAH_00010210*) was not differentially expressed in mature oocysts of *E. acervulina,* unlike *EMWEY_00036760*, which was differentially expressed in mature oocysts, as was its homolog in *E. acervulina* (*EAH_00022470*; >3.7 log_2_ FC in mature vs. immature oocysts but not a mature oocyst biomarker). In strain APU-2, one of the two genes identified (*EMWEY_00034320*) was also highly expressed at T0 (Appendix A). This gene, while >2-fold log_2_ differentially expressed at T36, was also constitutively expressed in each strain. Its homolog was previously identified as a mature oocyst biomarker in *E. acervulina* (*EAH_00057690*; >3-fold log_2_ upregulated). Therefore, this gene could be used to distinguish *E. maxima* APU-2 *mature* oocysts from immature oocysts, but its utility for other purposes may be limited.

### 3.8. Markers of Immaturity and Maturity Shared by Both Strains of Eimeria maxima

#### 3.8.1. Immature Oocysts

We then identified genes that demarcated markers of immaturity and maturity common to each strain. Given the strong temporal similarities in gene expression in the two strains, we sought to summarize which genes especially characterized either immature or mature oocysts of *Eimeria maxima.* To this end, we examined the 1871 significantly DEGs which contributed >1000 TPM at either hour 0 or 36 in each.

Only 13 of 1871 genes meeting these criteria were overexpressed (most >~2-fold log_2_) at hour 0 (Table 5). Four of these 13 were identified as constitutively expressedabove (*EMWEY_00025290, EMWEY_00029600, EMWEY_00045300, EMWEY_00057080*). Over half of the 13 genes encode hypothetical proteins; the others encode enzymes. For example, *EMWEY_00060390* (encoding phosphoserine aminotransferase) was expressed >7-fold log_2_ higher at T0 than at T36. Another gene, *EMWEY_00056350* (encoding subtilase) is homologous to a subtilase (*EAH_00048170*) we identified as upregulated in T0 oocysts in *E. acervulina.* Thus, this gene appears to strongly demarcate immature oocysts of both *E. acervulina* and *E. maxima.* None of the other genes matched a homolog in *E. acervulina* that contributed >1000 TPM and was differentially expressed in immature oocysts.

Except for one of the hypothetical protein-encoding genes (*EMWEY_00051470*), Blast2GO did not identify any informative hits from other Apicomplexa species (Appendix A). A total of 90 GO terms were identified for all genes and many were associated with the downregulated genes *EMWEY_00045300* (encoding MAP3K epsilon protein kinase) and *EMWEY_00025290* (isoleucyl-tRNA synthetase-related, related to CDS). *EMWEY_00060390* (phosphoserine aminotransferase), which exhibited elevated differential expression in both strains, was upregulated with at least ~5.7 log_2_ FC in *E. maxima* T0 oocysts. The associated GO terms pertained to serine and lysine biosynthesis.

The gene with the highest TPM at T0 in each strain (>26,000 TPM and at least ~1.9 log_2_ FC expressed vs. T36 oocysts) was *EMWEY_00057080* (a constitutively expressed gene described above). This was associated with the GO terms histone methylation and metal ion binding. *EMWEY_00004950* was upregulated >3-fold (log_2_) and encodes a hypothetical protein. It was associated with one GO term: hydrolase activity. *EMWEY_00018810* was ~3-fold log_2_ differentially expressed and associated with terms regarding mRNA processing and proteolysis.

*EMWEY_00051470* encodes a hypothetical protein that was expressed in immature oocysts ~5-fold log_2_ more than in each strain at hour 0. It resembles a wax ester synthase/acyl coenzyme A (acyl-CoA): diacylglycerol acyltransferase (WS/DGAT) with monoacylglycerol acyltransferase (MGAT) activity in *Cystoisospora suis*, associated with the GO terms O-acyltransferase activity and membrane. Four genes were associated with KEGG pathways: *EMWEY_00024360* (glycosylphosphatidylinositol (GPI)-anchor biosynthesis, purine metabolism), *EMWEY_00025290* (terpenoid backbone biosynthesis), *EMWEY_00045300* (human immunodeficiency virus 1 infection), and *EMWEY_00060390* (multiple amino acid, methane, and vitamin B6 metabolism).

#### 3.8.2. Mature Oocysts

We identified 32 significantly DEGs shared by strains at hour 36 when compared to initial conditions. Some of these achieved >14 log_2_ FC and contributed >10,000 TPM at hour 36. Interestingly, four genes were expressed strongly at both hours 0 and 36 (italicized in Table 5). The temporal directionality of expression differed. These four genes were highly constitutively expressed throughout sporulation.

Annotation beyond “hypothetical protein” was lacking for most of these 32 genes, including some exhibiting the highest temporal change in expression and contributing the most TPM. Blast2GO did find annotated homologs in some other Apicomplexans for some of these genes: 10/22 genes annotated as encoding hypothetical proteins had a more informative match (bolded in Appendix A). These were transmembrane proteins, microneme proteins, as well as some others. *EMWEY_00009150* (expressed >15,000 TPM in both strains and >10-fold log_2_ more at T36 than T0) was the highest expressed hypothetical protein-encoding gene that had a more informative BLAST hit (matched to *T. gondii* family D protein). This gene was also identified as a DEG differentiating strains at T48 vs. T36 (Appendix A). Sometimes, when a better annotation was identified, the best match (by BLAST e-value) was from *E. maxima*: these were for microneme proteins.

Overall, multiple clusters of genes were found, such as SAGs and micronemes. Along with a myosin protein, these types of protein-encoding genes were also identified in mature *E. acervulina* oocysts in our previous work. In fact, one-third (11/32 genes) bore strong homology to genes in *E. acervulina* previously identified as significantly differentially expressed biomarkers in mature oocysts (red in Appendix A). Thus, these 11 genes appear to represent especially important markers of maturity in the genus. Six other genes with this expression profile did not identifiably resemble a homolog in *E. acervulina,* seemingly representing markers of maturity in *E. maxima*.

For the 32 genes overexpressed at hour 36 in each strain of *E. maxima*, ~100 total GO terms were identified. Most of the GO terms were assigned to a few genes, such as *EMWEY_00025290* and *EMWEY_00045300.* Some genes undergoing strong upregulation at hour 36 had no associated GO terms or simply “membrane”, but a few such as *EMWEY_00014920*, *EMWEY_00017740*, *EMWEY_00029270*, and *EMWEY_00036540* had multiple terms and at least >8 fold log_2_ higher expression than T0 in both strains. Genes *EMWEY_00017740* and *EMWEY_00029270* were originally annotated to encode hypothetical proteins; Blast2GO found that the latter resembled a microneme protein in *E. maxima*. The only genes with associated KEGG pathways were the two in common (*EMWEY_00025290, EMWEY_00045300*, although upregulated in mature oocysts) with the T0 genes.

### 3.9. RT-qPCR-Validated RNA-Seq Data

To validate the estimates of gene expression from RNA-Seq, we selected three genes for confirmatory analysis using RT-qPCR. Three time points during sporulation were analyzed that captured expression at immature (T0), intermediate (T18), and mature (T36) stages of oocyst development. Two targeted genes were highly constitutively expressed with >1000 TPM at each time point (*EMWEY_0006260*, *EMWEY_00042090*, Appendix A) in each strain. Another gene, *EMWEY_00005470*, was ~3-fold log_2_ more expressed overall in APU-2 than in APU-1 with a greater expression vs. APU-1 at multiple time points (Table 2 and Table 4). We used these assays to compare the expression at hours 18 and 36 with expression at hour 0, our baseline. For these three genes, we compared log_2_ expression as estimated using RNA-Seq and RT-qPCR. As shown in Figure 7A, the expression levels of individual genes determined using RT-qPCR were consistent with those obtained using RNA-Seq.

As another measure to show qPCR-validated RNA-Seq as a method to investigate dynamic gene expression, data for the two methods (all three genes and time points, both strains) were correlated. These data correlated at a very high level (*r* = 0.996, Figure 7B) and the results are in line with our previous work studying expression during *E. acervulina* sporulation [37]. RT-qPCR accurately validates RNA-Seq expression according to various metrics.

## 4. Discussion

Coccidian infections cause enteric disease in wildlife, livestock, and people. Vaccines that incorporate multiple *Eimeria* species, and sometimes multiple strains of a single species, remain important to effective coccidiosis control programs, but success varies in part owing to the nuances of immunogenicity, strain variation, and fecundity, which impact treatment specificity and dosing [48,49]. *Eimeria maxima* can severely disrupt gut nutrient absorption, predisposing birds to microbial colonization and leading to serious complications (such as NE derived from superinfections with *Clostridium perfringens*) [7]. Diversity among strains of *E. maxima* complicates vaccination efforts because infection by one strain may provide little or no cross-protection against other strains [9,13,14,16,18,21,24,50].

Data from *Eimeria* provide important value for understanding concomitant processes in related parasites, such as *Cyclospora,* which compromises produce safety and causes human enteric disease. Unlike *Eimeria,* it is challenging to obtain large numbers of *Cyclospora* oocysts and difficult to achieve, much less study, their sporulation. We are unaware of studies characterizing transcriptomic changes in *Cyclospora* during sporulation (as we have undertaken here for *E. maxima*) or senescence, despite the value such studies might provide in risk assessment or risk reduction. Although we are beginning to appreciate the existence of genetic distinctions among *Cyclospora* isolates infecting humans, no data yet exist concerning differences in their gene expression or progression to the fully sporulated state.

To understand variation in *E. maxima*, researchers have characterized phenotypically different strains in terms of expressed antigens, omics, sporulation dynamics, pathogenesis/development in vivo, and fecundity [18,21,22,24,25,26,28,51,52]. Our previous work with two strains of *E. maxima* ([18,20]) showed that the APU-1 strain is more pathogenic than APU-2, exhibiting greater fecundity and host cell invasion. Here, we questioned whether differences in oocyst development might contribute to these differences. To this end, we built upon our previous work [37], which identified highly expressed genes in *E. acervulina* expressed throughout sporulation or most especially in mature or immature oocysts.

In this study, we found that *E. maxima* strains are similar in many respects, but some differences may underlie virulence phenotypes. We identified highly expressed genes throughout sporulation shared by both strains that have similarity to *E. acervulina* and other coccidia. Differential expression analysis between strains throughout sporulation identified potential markers of dissimilarity. Strains correlated highly at hour 36 but diverged at hour 48, and putative genes important to each time point were identified. In addition, we examined genes that may be specific to immature and mature oocysts of each strain, as well as genes that are shared by immature and mature oocysts in strains.

*E. maxima* oocysts highly express many genes throughout sporulation that are also expressed throughout the sporulation of *E. acervulina.* These include genes such as ATPases (cation-transporting), a zinc finger DHHC-containing protein, an oocyst wall protein, certain genes annotated to encode hypothetical proteins, and many modestly expressed genes that encode housekeeping functions (e.g., aquaporin, actin). The maximum mean expression of the constitutively expressed *E. maxima* genes occurred during hours 12 and 18 of sporulation, but their homologs in *E. acervulina* experienced peak expression at 4 h during sporulation. It is interesting that the latter species may have earlier expression during development, or perhaps sporulation is equivalent as *E. maxima* has a longer sporulation period. Genes important to sporulation in *E. tenella* were also reported to be expressed early during oocyst development (0–8 h) [36]. Therefore, critical processes to sporogony are activated early in multiple species of *Eimeria*. Further studies may be required to dissect what processes regulate the rate of development in each species. Some genes strongly expressed in *E. maxima* were not strongly expressed in *E. acervulina,* providing a basis to interrogate the elaboration of species-specific developmental determinants. One such gene (*EMWEY_00024920*) has GO terms pertaining to RNA and myosin dephosphorylation, metal ion binding, nucleus, and is associated with Programmed Death Ligand (PD-L1) expression and the PD-1 checkpoint pathway in cancer. Interestingly, its homolog in *E. acervulina* (*EAH_00006770*) was one we did not find to be constitutively expressed in that species; it was instead highly differentially expressed in immature vs. mature oocysts [37]. Three genes constitutively expressed in *E. maxima* have no homologs identified in *E. acervulina,* and the strongest expressed (*EMWEY_00057080*) encodes a hypothetical protein related to histone methylation and metal ion binding. This could be interesting to follow up further as an *E. maxima* viability gene.

We identified biomarkers capable of differentiating immature from mature *E. maxima*, among them many constitutively expressed genes. Relatively few of these demarcated strain APU-1 (and even fewer demarcated APU-2). Genes in immature APU-1 oocysts were largely downregulated, but several cell cycle kinase-encoding genes were upregulated. One upregulated gene (*EMWEY_00053520*) was homologous to an *E. acervulina* immature oocyst biomarker we previously identified. Another, *EMWEY_00052020* (encodes X-Pro-dipeptidyl-peptidase-domain containing protein), had the highest differential expression vs. APU-2 and may be a marker of APU-1 immature oocysts. Genes unique to mature APU-1 oocysts mostly comprised the constitutively expressed genes discussed above. However, *EMWEY_00006520* (encoding a protein similar to DNA-damage-inducible protein P from *E. coli*) was remarkable as the highest DEG compared to immature oocysts, yet it was not significantly differentially expressed in *E. acervulina* mature oocysts. *EMWEY_00036760* (encodes calmodulin) had a homolog in *E. acervulina* (*EAH_00022470)* that was highly differentially expressed in mature vs. immature oocysts of that species, however. For APU-2, the upregulated genes in immature oocysts included a group (ubiquitin, 40S ribosomal protein, ATP synthase beta chain) that may be important for oocyst developmental processes (respiration, protein synthesis). Genes important to mature APU-2 oocysts consisted of *EMWEY_00026670* and *EMWEY_00034320*, which encode an acid phosphatase and profilin protein, respectively. *EMWEY_00034320*, while also constitutively expressed in each strain, had a homolog in *E. acervulina* that was significantly differentially expressed in mature vs. immature oocysts. Therefore, it may represent a marker of APU-2 oocysts, but it can be used as a potential viability gene marker as well. Profilins bind actin, along with myosins, formins, and adhesins; they are part of the cellular apicomplexan glideosome essential for host cell invasion [53,54,55]. We previously identified types of glideosomal proteins as important in *E. acervulina* mature oocysts [37], and they likely serve similar functions in *E. maxima.*

Additionally, we examined the DEGs that demarcated immature and mature oocysts of *both* strains. Immature oocysts were associated with some constitutively expressed genes (*EMWEY_00025290* and *EMWEY_00057080* appeared especially high at time 0) and other enzymes such as phosphoserine aminotransfersase and subtilase. Certain highly expressed subtilases may demarcate immature oocysts, as the one identified in *E. maxima* (*EMWEY_00056350*) is homologous to a highly expressed subtilase in *E. acervulina* (*EAH_00048170*) that was significantly differentially expressed vs. mature oocysts. In general, GO terms and KEGG pathways identified may define functions particularly important at the inception of sporulation, such as GPI-anchor biosynthesis, histone methylation, and mRNA processing and proteolysis. A larger group of genes was identified in mature *E. maxima* oocysts. Several of these genes had exceptional fold changes compared to immature oocysts. While some of these genes were constitutively expressed throughout sporulation, we identified groups of genes that previously appeared to be gene markers of maturity in *E. acervulina* [37]. These included genes encoding SAGs, transmembrane proteins, microneme proteins, and myosin. In addition to our previous work, the work of others has shown the importance of these types of encoding genes to different stages of *Eimeria* spp. [26,28,35,51,56,57], including characteristic expression in sporulating oocysts or sporozoites.

We also identified genes that may characterize either strain, throughout sporulation and also at specific intervals. We found that the DEGs identified in our overall analysis were also differentially expressed at particular time points, such as hour 36 (when the transcription between two strains correlated highly). Throughout sporulation, a relatively small number of DEGs (~20 of over 6000 annotated genes) distinguished the strains according to our selection criteria. Genes upregulated in APU-1 mostly encoded hypothetical proteins, but one (*EMWEY_00023530)* encodes the 18kDa cyclophilin A. Cyclophilins catalyze protein isomerization and are involved in proper protein folding; they are critical for pathogen infection [58]. There are multiple cyclophilin families in apicomplexans and these may have roles in RNA processing and act as chaperones [59,60]. Cyclophilin A is a receptor for the immunosuppressive drug cyclosporin A, which inhibits infection or replication of many apicomplexan parasites, including *P. falciparum*, *T. gondii*, and *Eimeria* spp. [60]. The increased differential expression of cyclosporin A in APU-1 vs. APU-2 indicates this critical protein is especially important in the development of the virulent strain, yet we cannot exactly attribute its function for virulence.

Some additional APU-1 DEGs were identified at hour 36 that differed from our overall analysis and may also be important for separating strains. *EMWEY_00006520* (aforementioned), *EMWEY_00017660* (splicing factor 3B subunit), and *EMWEY_00052370* (uroporphyrinogen decarboxylase) were on the borderline of differential expression but notable for their potential roles relating to development and DNA binding/damage repair, spliceosome assembly, and heme biosynthetic process, respectively. Additionally, genes annotated as encoding hypothetical proteins (*EMWEY_00002130*, *EMWEY_00005570,* and *EMWEY_00056530*) were upregulated >2-fold (log_2_) in APU-1 and associated with GO terms pertaining to DNA-binding/transcription and protein binding.

Genes significantly upregulated in APU-2 in our overall analysis had more functional annotations (although we could not glean much information for the most upregulated transcript, *EMWEY_0008090*). They were mostly associated with the GO term membrane, but genes *EMWEY_00036080* (thioredoxin), *EMWEY_00039050* (OTU-like cysteine protease domain-containing protein), and *EMWEY_00058460* each bear multiple terms pertaining to protein processes (e.g., -folding, deubiquination, export, binding, assembly). Thioredoxins (with roles in oxidative stress response) and OTU-like cysteine proteases (deubiquinating family enzymes) regulate cellular processes in apicomplexans and have key roles in cellular proliferation and apicoplast import [61,62,63,64]. *EMWEY_00058460* (encodes EF-1α) appeared especially interesting since it had >6 log_2_ FC, encompassed many of the GO terms, and had multiple linked KEGG pathways. EF-1α is a GTPase that transfers aminoacylated tRNAs to ribosomes during protein synthesis. In Apicomplexa, it is a reactive antigen expressed on the apical region of sporozoites, and it is important for invasion. It is currently being studied as a vaccine candidate in multiple Apicomplexan species, associated with robust anti-EF-1α responses [65,66,67,68]. In addition, multiple SAG genes were upregulated in APU-2, and these were different from those we found in common to mature *E. maxima* oocysts.

As with the APU-1 strain, some additional upregulated DEGs were identified at hour 36 in APU-2. Genes *EMWEY_00000860* (20 kDa cyclophilin)*, EMWEY_00011930* (heat shock protein)*, EMWEY_00052300* (NAC domain-containing protein), *EMWEY_00054680* (aspartate aminotransferase), and *EMWEY_00059760* (asparaginase) were notable for their higher differential expression and/or GO term annotations. These terms included protein isomerization/folding, response to misfolded proteins, translation, aspartate amino acid biosynthesis and pyridoxal phosphate binding, and asparaginase activity.

Taken together, two cyclophilin-encoding genes appear to be important during the development of APU-2 (upregulated and downregulated) and a heat shock protein was also upregulated in APU-2. This may reinforce the theme of proper protein folding and chaperone activity as especially important to the less virulent strain, perhaps temporally dependent. It appeared that amino acid metabolism was upregulated in APU-2 as well (more so at 36 h). Hypothetical proteins with expression on the borderline but with GO terms included *EMWEY_00017480* and *EMWEY_00056980*, whose GO terms pertained to regulation of transcription, nucleus, and membrane. These genes therefore appeared to have similar functionality important to multiple forms of transcriptional processes. Genes in both *E. maxima* strains at T36 generally had higher TPM at this time in *E. maxima* than their homologs in mature *E. acervulina* oocysts [37]. However, homologs of the hypothetical protein-encoding genes *EMWEY_00031340* and *EMWEY_00056980* (upregulated in APU-2) were expressed to a much greater extent in *E. acervulina*. In fact, the homologs of these two genes in *E. acervulina*, as well as *EMWEY_00011930*, were significantly differentially expressed in mature vs. immature oocysts [37], perhaps indicating important similarities in mature oocysts between species (we did not perform a cross-species differential expression analysis).

Based on transcription, the two strains agreed highly at hour 36 and then diverged considerably at 48 h. We therefore examined whether DEGs at the different time points may vary, accounting for some of the transcriptional divergence. We found ~65% DEGs between strains at hour 36 were also differentially expressed at hour 48. Most of these genes were expressed with <100 TPM in each strain. The higher expressed genes in APU-1 were those we identified above as differentially expressed. For example, *EMWEY_00006520* was upregulated in APU-1 when comparing strains at hour 36, and it was uniquely differentially expressed compared to immature oocysts. Therefore, it may be a promising marker of mature APU-1 oocysts. Interestingly, its lack of differential expression between strains at hour 48 may indicate it is less important after oocysts have sporulated. A greater number of DEGs at hour 48 were not identified at hour 36. These genes also experienced higher transcription and a greater range in differential expression. Strain APU-2 upregulated several genes highly at T48 compared to APU-1, such as hypothetical proteins with homology to membrane and serine-rich proteins in other species. In the genes particular to hours 36 and 48, some commonalities emerged. We found that at both time points, genes were annotated that encode DNA repair, stress response, helicases, subtilases, translation initiation factors, and ATPase subunits. Therefore, it appeared that common suites of certain proteins are important for mature oocysts in general. At hour 48, oocysts appear to differentially express unique genes such as those encoding adaptin, chromosome segregation proteins, DNA polymerases, microneme proteins, and SAGs, among others.

Notwithstanding their notable virulence differences, we found that the two strains of *E. maxima* nonetheless undergo decidedly similar transcriptional changes as they sporulate. During the first 12 h, contemporaneous measurements affirm strongly correlated transcriptomes. Then, the less virulent strain (APU-2) transiently outpaces the more virulent strain. During hours 18–24, microscopy indicated more sporulated APU-2 oocysts. These data affirmed that each strain reaches maximal sporulation by about 30 h, at which time their transcriptional profiles match especially closely. This echoes prior reports, from microscopy, tracking the timing of sporulation in *E. maxima* (Edgar, 1955; Norton and Chard, 1983). Interestingly, the transcriptional profiles thereafter begin to diverge, characterized by weaker correlation at 48 h, and more differentially expressed genes. Such divergence may prefigure or even contribute to what distinguishes each strain once it infects the next host. Thus, the genes identified as diverging in their expression by hour 48 deserve future examination for any functional role they may play in virulence differences among the strains.

Selection for early oocyst maturation can be accomplished via experimental passage through chickens, producing attenuated (less pathogenic) strains of *Eimeria* that are also faster to develop (precocious). In that light, it is interesting that the less pathogenic, naturally occurring strain studied here (APU-2) progressed though sporulation somewhat faster. Precocious *Eimeria* strains are less pathogenic and exhibit reduced fecundity but have high immunogenicity [12,69,70,71,72]. They exhibit a different life cycle (often with faster sporozoite invasion) compared to the parent or virulent wild-type strain, which likely accounts for their lower reproductive potential. Our findings indicate that APU-2, although naturally occurring, exhibits life history characteristics akin to those produced via artificial selection during experimental passage. Broader sampling would be needed to discern whether virulence differences broadly entail trade-offs with the pace of sporulation.

It is tempting to speculate on the involvement of the DEGs we have identified in relation to the virulence phenotypes of the *E. maxima* strains. Unfortunately, some of the more interesting genes often escaped functional characterization efforts, limiting our conclusions, and necessitating further work. Groups of genes in APU-2 may indicate the less virulent strain upregulates more developmental and sporozoite or invasion genes but some genes with similar functions (e.g., SAGs, micronemes, myosin, membrane proteins) were also found to be in common to strains at hour 36 or 48. SAG genes were differentially expressed in APU-2 oocysts with elevated expression at 36 h and different upregulated SAGs were shared by both strains. The relevance of higher expressed SAGs in one strain vs. another is unclear. Whether these contribute to differences in the rate of sporulation, or the degree of virulence expressed in the host, remains unclear. SAGs are principal surface antigens of invasive Apicomplexan stages, involved in pathogenicity, maturation, and other processes [38,73,74]. We found previously, using RNA-Seq, that SAGs were differentially expressed in mature vs. immature oocysts in *E. acervulina* [37]. Hu et al. (2018) reported that precocious, sporulated *E. maxima* oocysts upregulate translation initiation factors, transcription initiation factors, and organelle invasion proteins (SAGs, micronemes, calcium-dependent protein kinases, myosins) that could be used in invasion, proliferation, and division processes [28]. It appeared that sporulated oocysts of both the virulent and precocious strains exhibited highly transcribed organelle genes, however, which is consistent with oocyst sporulation (sporozoite differentiation, organelle emergence). Two gene expression studies in *E. tenella* by Matsubayashi et al. [56,57] found genes relating to glycolysis, TCA, and the pentose phosphate pathway were upregulated in sporulating oocysts of a virulent strain, whereas virulent sporozoites had upregulated carbohydrate metabolism and attachment genes. Also, secreted apical complex proteins, proteases, mitochondrial proteins, and transporters were strongly upregulated in virulent sporozoites, whereas the precocious strain has upregulated genes associated with cell survival, development, and proliferation. The authors theorized that perhaps the parent strains survive long before the invasion and invade actively/successfully into host cells, whereas proliferative processes influence precocity. Our work contrasts with this study in that virulent APU-1 did not exclusively express these types of molecules.

Considering prior studies, it is more difficult to ascribe other genes to certain other phenotypes. Dong et al. [25] reported that a precocious *E. maxima* strain (sporulated oocysts) upregulates (vs. a parental, virulent strain) a rhomboid family domain-containing protein (possibly for invasion, faster schizogonic development) and downregulates a serpin (serine protease inhibitor), a cation-transporting ATPase, heat-shock protein 70, and a transhydrogenase. In our studies, we found both *E. maxima* strains expressed a rhomboid family domain-containing protein, two cation-transporting ATPases, and a transhydrogenase consistently throughout sporulation (the *E. acervulina* homologs were also constitutively expressed in our previous RNA-Seq). While we noted a higher expression in mature vs. immature oocysts for these genes, our differential expression analysis found that one of the cation-transporting ATPases (*EMWEY_00006260*) and the transhydrogenase (*EMWEY_00011310*) were only upregulated in mature APU-1 oocysts. The downregulation of other genes may correspond to adaptive changes during acquisition of the precocious phenotype. The upregulation of the cation-transporting ATPase and a calmodulin in mature APU-1 oocysts may indicate the increased importance of calcium regulation in virulent strains. Apicomplexans employ a variety of calcium-modulating proteins for various roles such as motility and differentiation [75].

Hu et al. [28] also reported that some genes we described above (as constitutively expressed and more expressed in APU-1 *mature* oocysts) were upregulated in precocious vs. parental unsporulated oocysts. These included genes such as aspartyl protease, insulysin, a rhomboid family domain-containing protein, cation-transporting ATPase, a zinc finger DHHC domain-containing proteins, among others. In addition to that described above, we also found a zinc finger DHHC domain-containing protein (*EMWEY_00000500*), an insulysin (*EMWEY_00013380*), and aspartyl protease (*EMWEY_00013390)* were constitutively expressed in *E. maxima* strains. Again, these genes (except for *EMWEY_00013390*) were significantly upregulated in mature APU-1 oocysts. Interestingly, the constitutively expressed genes discussed here were not found to be differentially expressed in *E. acervulina* oocysts in our earlier data.

Addressing the animal health and food safety challenges posed by coccidian oocysts requires a global understanding of how oocysts mature. Despite their phenotypic differences, we found highly parallel development in two strains of *E. maxima* and that genes expressed through their sporulation largely parallel those expressed throughout sporulation in another species, *E. acervulina.* Constitutively expressed genes during sporulation have homologs in other types of coccidia, including *Cyclospora*. These data therefore contribute to broader inter-specific (and inter-generic) comparisons, bringing into focus what uniquely characterizes each species’ development and what processes generally mediate coccidian oocyst development. Thus, such data build a foundation to understand coccidian development *writ large*, which may find broad application in animal health and food safety. For example, transcripts enabling better assessment of *Eimeria* oocyst viability may help the produce industry verify that a given vaccine dose remains potent. Likewise, food safety professionals could better assess and manage risks if they could determine, in vitro, the viability of *Cyclospora* contaminating water or produce. To these ends, our work provides the first cross-species analysis of the degree of conservation among constitutively expressed genes throughout sporulation in any two species of coccidia.

By identifying genes important throughout sporulation of *E. maxima* and *E. acervulina*, or important at stages in their maturation, comparative transcriptomics now establishes testable hypotheses in other species, such as *Eimeria tenella* and *Cyclospora cayetanensis.* Genes elaborated to an especially great extent in mature oocysts also form a group that may prove important during subsequent developmental stages, particularly in sporozoites as they invade host cells and trophozoites replicating in the host. Such studies would need to discern, for such intracellular parasitic stages, transcriptional processes derived from both the parasite and the host cell. Furthermore, characterizing gene expression in maturing and senescing oocysts holds the promise of further improving the in vitro assessment of oocyst viability.

These data helpfully constrain targets for characterizing *Cyclospora* maturation. Precisely replicating this study might demand unachievably numerous and synchronized cohorts of *Cyclospora* (derived, as they are, not from experimental animals but rather from clinical patients). Nonetheless, advances in single-cell transcriptomics might enable the evaluation of sparser RNA templates, enabling empirical evaluation of the predictive value of surrogate data in understanding, and better managing, the risks such organisms pose to human health.

## Figures and Tables

**Figure 1 pathogens-13-00002-f001:**
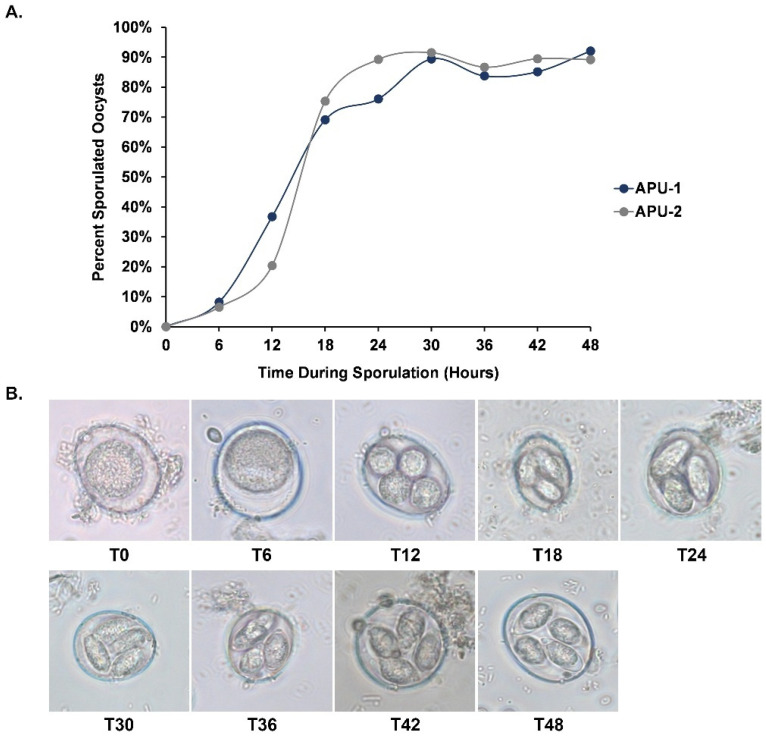
Differential oocyst development in *E. maxima* strains APU-1 and APU-2. (**A**). Percent sporulation of strains during oocyst development. Strain APU-2 reached maximum sporulation sooner. Strains reached maximum sporulation at 30 h. (**B**). Representative photomicrographs of developing *E. maxima* oocysts at each time point. Additional time points T30 and T42 are included. Unsporulated oocysts (T0) are diploid and go through meiosis. Four haploid sporoblasts are visible by 12 h (T12) and these become sporocysts (T18–T48). Two sporozoites form in each for the four sporocysts.

**Figure 2 pathogens-13-00002-f002:**
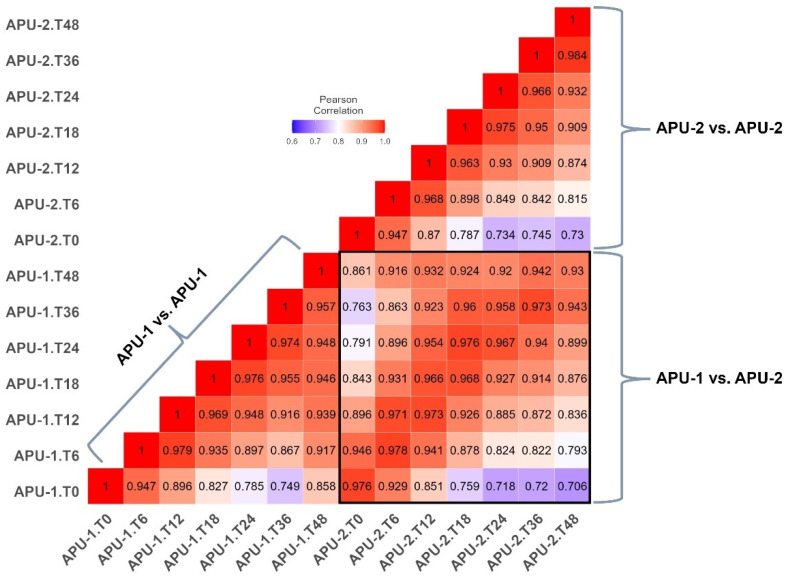
Pearson’s correlation matrix of transcription between strains APU-1 and APU-2. Mean log_2_ of mapped reads from three replicates per time point were compared for each strain every 6–12 h during oocyst sporulation. Individual squares represent Pearson’s product correlations between (and within) strains at each time point. Correlations generally decreased with time in each strain. The boxed in area focuses on APU-1 vs. APU-2 correlations, specifically. Note that contemporaneous correlations were generally strongest at the same time point, with exceptions.

**Figure 3 pathogens-13-00002-f003:**
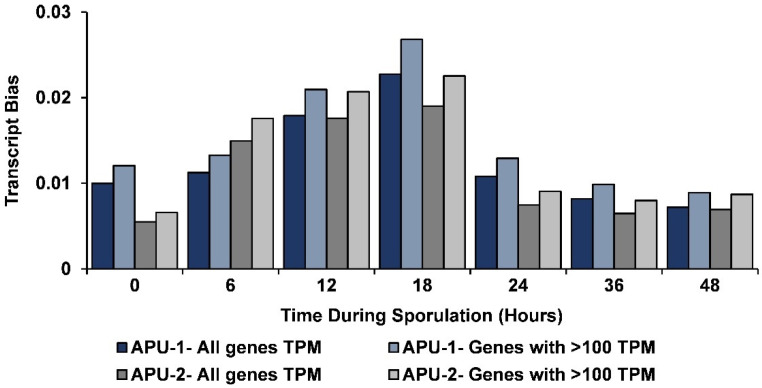
Transcript bias during oocyst sporulation in *E. maxima* strains APU-1 and APU-2. The bias in transcription (when a subset of genes most disproportionately contribute to transcripts) is greatest mid-sporulation and peaks at hour 18 (T18). Parallel temporal patterns in transcriptional bias hold whether the analysis is performed on all genes (black, APU-1; dark gray, APU-2) or restricted to those each contributing at least 100 transcripts per million (>100 TPM) (blue, APU-1; gray, APU-2). Transcriptional bias was calculated at each time point (based on mean TPM from three replicates per time point) for each strain.

**Figure 4 pathogens-13-00002-f004:**
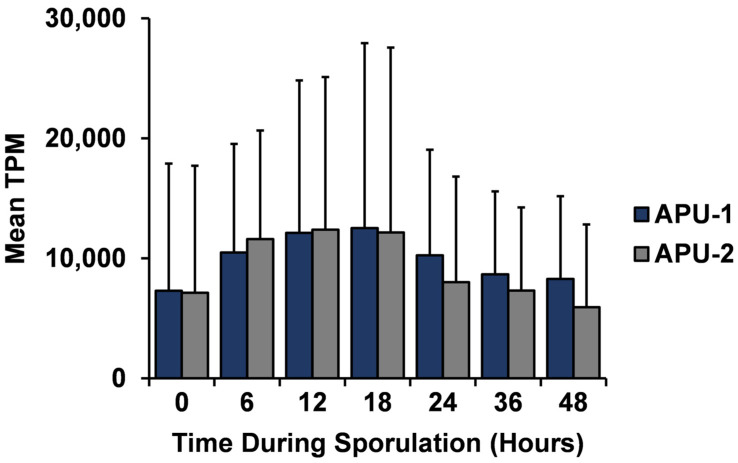
Temporal trend of constitutively expressed genes in *E. maxima* strains APU-1 and APU-2. Mean TPM of these 58 genes is displayed over the time course of sporulation. Their mean TPM peaked at 12–18 h. These data support the conclusion that most transcripts important to the continuous development of one strain are also important to the other strain.

**Figure 5 pathogens-13-00002-f005:**
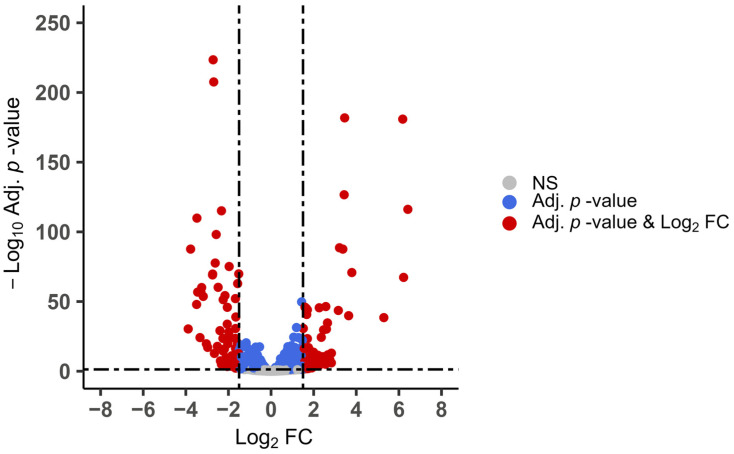
Differentially expressed genes (DEGs) during sporulation in *E. maxima* strains APU-1 and APU-2. Expression during sporulation was compared between strains using DESeq2. Threshold criteria of >1.5 or <−1.5 log_2_ fold change (FC) with adjusted *p* < 0.05 (Adj. *p*-value) were utilized. By these criteria, 204 significantly DEGs were identified (in red). This plot depicts expression of *E. maxima* APU-2 vs. APU-1 (143 upregulated, 61 downregulated genes in APU-2). NS indicates non-significant.

**Figure 6 pathogens-13-00002-f006:**
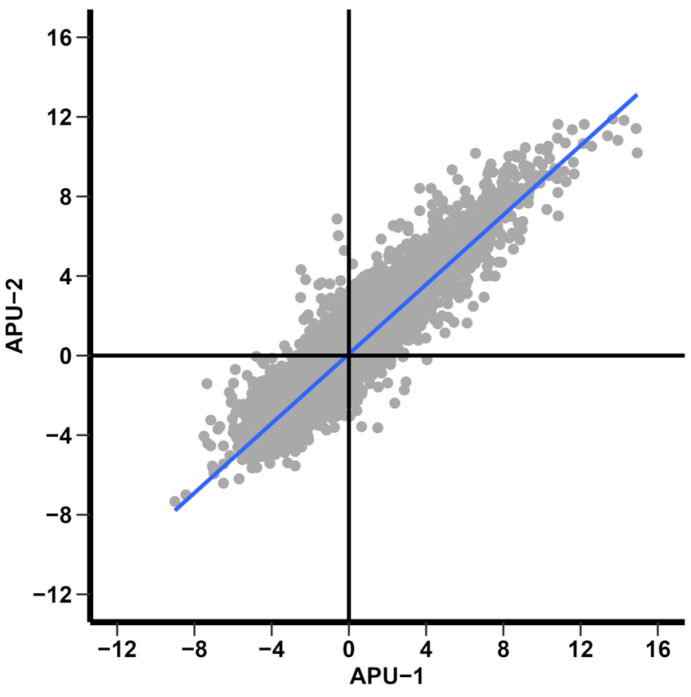
Log_2_ FC differential expression between time points T36 and T0 for strains APU-1 and APU-2. Much directional similarity between expressed genes is evident, concordance in top-right and bottom-left quadrants. The blue line indicates the regression line. Note: for 75 of 6057 annotated genes, differential expression could not be determined between time points in strains; therefore, the graph depicts differential expression of 5982 genes.

**Figure 7 pathogens-13-00002-f007:**
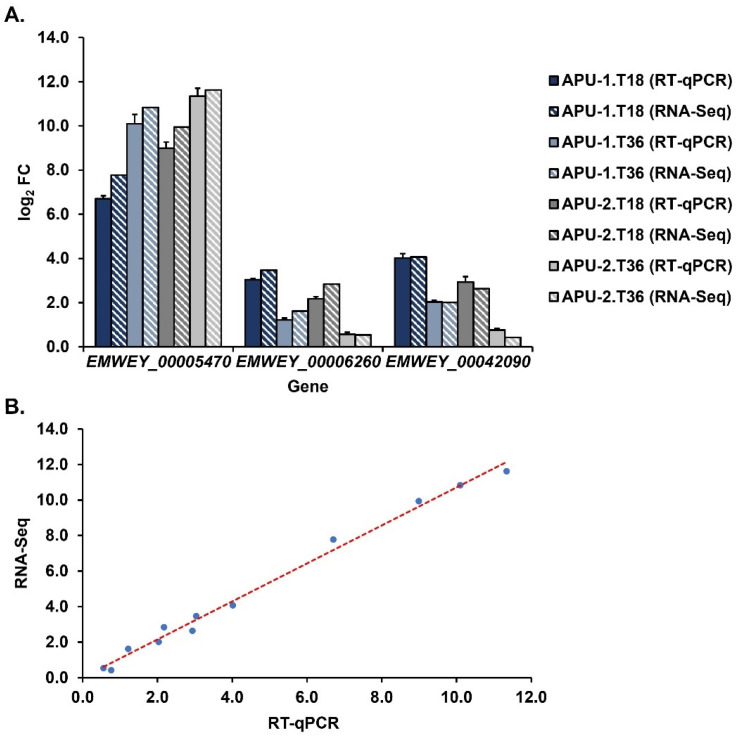
Relative expression of selected genes correlates highly between RNA-Seq and RT-qPCR assays. (**A**). Temporal changes in expression of genes selected for validation using RT-qPCR at time points T18 and T36 during oocyst sporulation. The log_2_ FC using RT-qPCR and RNA-Seq (differential expression by DESeq2) are compared. Gene expression at each time point is depicted relative to initial transcription level (T0). Bars with solid fill represent RT-qPCR and bars with stripe fill represent RNA-Seq. Error bars depict variance among RT-qPCR estimates from three replicate experiments that each employed three replicates per time point. (**B**). Correlation plot of RT-qPCR vs. RNA-Seq log_2_ FC data shown in (**A**). This incorporates data from all three genes and time points T18, T36 (both strains). Red dashes indicate regression line.

**Table 1 pathogens-13-00002-t001:** RNA-Seq expression analytics for *E. maxima* strains APU-1 and APU-2 during sporulation.

**Time** **(h)**	**Most Transcribed APU-1 Gene (TPM)**	**Number of APU-1 Genes**				
		**TPM > 1000**	**TPM > 100**	**TPM < 100**	**Upregulated ***	**Downregulated ***
0	64,395	127	1332	4725	N/A	N/A
6	29,934	99	970	5087	372	656
12	59,203	83	774	5283	690	695
18	60,631	74	723	5334	1061	1089
24	31,484	102	1000	5057	1168	1011
36	25,751	128	1061	4996	1335	1105
48	29,566	128	1207	4850	1192	283
**Time** **(h)**	**Most Transcribed APU-2 Gene (TPM)**	**Number of APU-2 genes**				
		**TPM > 1000**	**TPM > 100**	**TPM < 100**	**Upregulated ***	**Downregulated ***
0	30,940	135	1381	4676	N/A	N/A
6	36,047	86	847	5210	491	436
12	39,524	74	718	5339	875	986
18	48,731	75	745	5312	1168	1070
24	26,560	139	1080	4977	1393	1092
36	26,833	134	1146	4911	1456	1043
48	48,082	137	1231	4826	1591	1087

TPM = Transcripts Per Million, averaged among biological replicates, for each time point during sporulation. * Genes expressed significantly more, or less, than initially (T0); pairwise differential expression thresholds of >1.5 or <−1.5 log_2_ fold change (FC) and adjusted *p*-value < 0.05.

**Table 2 pathogens-13-00002-t002:** Significantly differentially expressed genes (DEGs) between APU-1 and APU-2 during sporulation.

			APU-1 TPM				APU-2 TPM				
	Gene EMWEY_	Description	T0	T18	T36	Mean TPM ***	T0	T18	T36	Mean TPM ***	Log_2_ FC
**APU-1 DEGS ***	01070	hyp. prot., conserved CDS	187.81	76.96	93.74	108.49	65.82	24.93	39.46	36.26	1.57
	*07700*	*hyp. prot., conserved CDS*	*9.67*	*378.89*	*1066.08*	*560.44*	*5.91*	*118.66*	*377.62*	*151.32*	*1.59*
	12880	hyp. prot., conserved CDS	6.49	66.56	241.64	105.98	8.60	21.26	63.37	30.92	1.60
	23520	hyp. prot., conserved CDS	24.52	149.62	271.28	144.37	54.38	28.59	42.33	39.33	1.85
	*23530*	*18 kDa cyclophilin, putative CDS*	*2362.40*	*2058.24*	*1688.28*	*1898.51*	*648.83*	*472.21*	*443.83*	*435.89*	*2.06*
	33050	hyp. prot., conserved CDS	166.11	92.34	129.99	132.67	32.97	20.49	41.92	35.15	1.97
	41620	hyp. prot., conserved CDS	306.79	172.87	212.61	223.12	42.17	23.97	42.12	33.11	2.72
	48910	hyp. prot., conserved CDS	473.70	393.02	420.48	411.29	68.62	90.36	102.65	81.01	2.25
	56530	hyp. prot., conserved CDS	95.22	64.43	291.27	145.30	15.11	7.23	23.56	13.98	3.29
	57350	hyp. prot. CDS	313.42	50.66	28.68	109.89	47.00	9.34	11.21	27.49	1.63
**APU-2 DEGS ****	05470	SAG family member CDS	0.17	17.57	214.75	62.90	1.10	519.82	3275.43	1129.06	2.83
	*07700*	*hyp. prot., conserved CDS*	*9.67*	*378.89*	*1066.08*	*560.44*	*5.91*	*118.66*	*377.62*	*151.32*	*−1.59*
	08090	hyp. prot. CDS	0.38	1.49	0.91	0.89	30.28	99.91	129.20	105.36	6.42
	*23530*	*18 kDa cyclophilin, putative CDS*	*2362.40*	*2058.24*	*1688.28*	*1898.51*	*648.83*	*472.21*	*443.83*	*435.89*	*−2.06*
	36080	thioredoxin, putative CDS	42.95	11.73	43.11	28.25	54.98	10.79	114.84	108.85	1.53
	37810	hyp. prot., conserved CDS	0.23	1.03	61.97	18.61	0.33	5.24	199.46	106.67	1.51
	39050	OTU-like cysteine protease domain-containing protein, putative CDS	12.05	8.02	9.62	10.08	162.31	71.03	120.38	110.99	3.45
	42800	hyp. prot., conserved CDS	2.44	2.30	69.55	21.28	3.65	14.70	200.55	101.14	1.56
	58460	elongation factor 1-alpha, putative	5.94	0.31	0.46	1.54	207.69	38.89	106.73	150.15	6.22
	59740	SAG family member CDS	0.55	34.95	135.31	57.19	11.59	266.61	789.42	371.32	2.07

TPM = Transcripts Per Million, averaged among replicates, for different developmental time points T0, T18, and T36. hyp. prot. = hypothetical protein. Italicized genes (*n* = *2*) are shared and significantly differentially expressed between strains with >100 overall mean TPM in each strain. * APU-1 differentially expressed genes (DEGs) expressed significantly more in APU-1 vs. APU-2 DESeq overall comparison, meeting thresholds of >1.5 or <−1.5 log_2_ fold change (FC) and adjusted *p* < 0.05 with >100 overall mean TPM in APU-1. ** APU-2 differentially expressed genes (DEGs) expressed significantly more in APU-1 vs. APU-2 DESeq overall comparison, meeting thresholds of >1.5 or <−1.5 log_2_ fold change (FC) and adjusted *p* < 0.05 with >100 overall mean TPM in APU-2. *** Mean TPM = overall mean gene TPM for all replicates and time points throughout sporulation.

**Table 3 pathogens-13-00002-t003:** Summary of differential expression comparing oocyst development (hours) between *E. maxima* strains.

APU-1 Time (h)	APU-2 Time (h)	Total DEGs *	Correlation **
6	6	216	0.978
24	18	207	0.976
0	0	201	0.976
12	12	229	0.973
36	36	284	0.973
18	18	399	0.968
24	24	466	0.967
36	48	975	0.943
48	36	551	0.942
48	48	850	0.930
18	24	1389	0.927

* Differentially expressed genes (DEGs)—genes expressed significantly more or less between strains; pairwise differential expression thresholds of >1.5 or <−1.5 log_2_ fold change (FC) and adjusted *p* < 0.05. ** Correlation of the two strains, based on the mean log_2_ of mapped reads for three replicates per time point (Figure 2). Green rows denote concurrent comparisons. Yellow rows depict strain-to-strain correlations at different times. Entries are listed in decreasing order of correlation.

**Table 4 pathogens-13-00002-t004:** Significantly differentially expressed genes with elevated expression between APU-1 and APU-2 at T36.

	Gene EMWEY_	Description	APU-1.T36 Mean TPM ***	APU-2.T36 Mean TPM ***	Log_2_ FC
**APU-1 DEGs ***	02130	hyp. prot., conserved CDS	328.64	89.27	2.12
	*05470*	*SAG family member CDS*	*214.75*	*3275.43*	*−3.58*
	05570	hyp. prot., conserved CDS	121.20	17.02	3.06
	*06520*	*Similar to DNA-damage-inducible protein P from E. coli, related CDS*	*1425.44*	*524.72*	*1.70*
	*07700*	*hyp. prot., conserved CDS*	*1066.08*	*377.62*	*1.71*
	07780	hyp. prot. CDS	128.54	47.50	1.67
	12880	hyp. prot., conserved CDS	241.64	63.37	2.18
	14910	hyp. prot., conserved CDS	168.53	69.86	1.53
	*17480*	*hyp. prot. CDS*	*189.83*	*751.55*	*−1.60*
	17660	splicing factor 3B subunit 1, putative CDS	158.71	58.98	1.69
	*21420*	*hyp. prot., conserved CDS*	*424.56*	*165.62*	*1.61*
	23520	hyp. prot., conserved CDS	271.28	42.33	2.93
	*23530*	*18 kDa cyclophilin, putative CDS*	*1688.28*	*443.83*	*2.13*
	*31340*	*hyp. prot., conserved CDS*	*246.23*	*930.80*	*−1.52*
	33050	hyp. prot., conserved CDS	129.99	41.92	1.90
	41620	hyp. prot., conserved CDS	212.61	42.12	2.48
	*48910*	*hyp. prot., conserved CDS*	*420.48*	*102.65*	*2.26*
	52370	uroporphyrinogen decarboxylase, putative CDS	111.94	42.78	1.62
	56530	hyp. prot., conserved CDS	291.27	23.56	3.85
	*56980*	*hyp. prot., conserved CDS*	*120.87*	*501.68*	*−1.66*
	*59740*	*SAG family member CDS*	*135.31*	*789.42*	*−2.20*
**APU-2 DEGs ****	00860	20 kDa cyclophilin precursor, putative CDS	16.91	127.16	2.57
	*05470*	*SAG family member CDS*	*214.75*	*3275.43*	*3.58*
	*06520*	*Similar to DNA-damage-inducible protein P from E. coli, related CDS*	*1425.44*	*524.72*	*−1.70*
	07270	hyp. prot., conserved CDS	27.42	108.51	1.61
	*07700*	*hyp. prot., conserved CDS*	*1066.08*	*377.62*	*−1.71*
	08090	hyp. prot. CDS	0.91	129.20	6.35
	10050	hyp. prot. CDS	25.17	141.14	2.10
	11930	heat shock protein, related CDS	33.00	137.23	1.64
	*17480*	*hyp. prot. CDS*	*189.83*	*751.55*	*1.60*
	*21420*	*hyp. prot., conserved CDS*	*424.56*	*165.62*	*−1.61*
	*23530*	*18 kDa cyclophilin, putative CDS*	*1688.28*	*443.83*	*−2.13*
	*31340*	*hyp. prot., conserved CDS*	*246.23*	*930.80*	*1.52*
	39050	OTU-like cysteine protease domain-containing protein, putative CDS	9.62	120.38	3.20
	*48910*	*hyp. prot., conserved CDS*	*420.48*	*102.65*	*−2.26*
	52050	hyp. prot., conserved CDS	12.26	103.61	2.61
	52300	NAC domain containing protein, putative CDS	28.95	104.44	1.51
	54680	aspartate aminotransferase, putative CDS	32.51	199.36	2.19
	*56980*	*hyp. prot., conserved CDS*	*120.87*	*501.68*	*1.66*
	58460	elongation factor 1-alpha, putative CDS	0.46	106.73	6.72
	*59740*	*SAG family member CDS*	*135.31*	*789.42*	*2.20*
	59760	asparaginase, putative CDS	37.78	144.77	1.51

hyp. prot. = hypothetical protein. Italicized genes (*n* = 10) are shared and significantly differentially expressed between strains at hour 36 with mean TPM > 100 at hour 36 in each strain. * Differentially expressed genes (DEGs) expressed significantly more in APU-1 vs. APU-2 DESeq pairwise comparison, meeting thresholds of >1.5 or <−1.5 log_2_ fold change (FC) and adjusted *p* < 0.05 with >100 overall mean TPM in APU-1 at T36. ** Differentially expressed genes (DEGs) expressed significantly more in APU-1 vs. APU-2 DESeq pairwise comparison, meeting thresholds of >1.5 or <−1.5 log_2_ fold change (FC) and adjusted *p* < 0.05 with >100 overall mean TPM in APU-2 at T36. *** Mean TPM = Transcripts Per Million, averaged among T36 replicates in each strain.

**Table 5 pathogens-13-00002-t005:** Significantly differentially expressed genes (DEGs) between immature (T0) and mature (T36) oocysts in common to both *E. maxima* strains.

	Gene EMWEY_	Description	APU-1.T0 Mean TPM **	APU-1.T36 Mean TPM **	APU-1 Log_2_ FC	APU-2.T0 Mean TPM **	APU-2.T36 Mean TPM **	APU-2 Log_2_ FC **
**T0 DEGs ***	00690	hyp. prot., conserved CDS	5123	1001.90	1.77	4302	903.57	2.06
	04950	hyp. prot., conserved CDS	1415	90.29	3.39	1191	65.20	3.99
	18810	hyp. prot., conserved CDS	1632	165.39	2.73	1173	147.22	2.67
	24360	N-acetylglucosaminylphosphatidylinositol deacetylase, putative CDS	1237	78.02	3.41	1572	73.30	4.22
	*25290*	*isoleucyl-tRNA synthetase-related, related CDS*	*49,929*	*5216.95*	*2.69*	*19,239*	*4363.92*	*1.80*
	*29600*	oocyst wall protein COWP, putative CDS	4769	16,820.59	−2.40	4278	15,947.87	−2.13
	41160	hyp. prot., conserved CDS	6441	149.43	4.86	4380	448.79	2.88
	43880	hyp. prot., conserved CDS	1043	164.31	2.09	1251	129.49	3.06
	*45300*	*MAP3K epsilon protein kinase, related CDS*	*1220*	*3758.14*	−*2.20*	*1045*	*3707.81*	−*2.05*
	51470	hyp. prot., conserved CDS	1974	30.18	5.45	1433	48.64	4.57
	56350	subtilase family serine protease, putative CDS	2983	413.98	2.28	2087	413.85	2.03
	*57080*	*hyp. prot., conserved CDS*	*64,395*	*6815.25*	*2.68*	*26,585*	*5727.77*	*1.87*
	60390	phosphoserine aminotransferase, putative CDS	2009	10.06	7.02	1253	18.41	5.69
**T36 DEGs ***	00710	hyp. prot., conserved CDS	394.06	1314	2.32	412.18	1031	1.56
	01470	hyp. prot. CDS	1.19	16,791	14.25	6.93	26,833	11.82
	05500	SAG family member CDS	8.07	1834	8.39	12.14	1451	7.05
	06190	hyp. prot., conserved CDS	21.10	1856	7.03	17.37	2930	7.55
	08040	SAG family member CDS	3.11	13,380	12.57	8.14	11,294	10.52
	09150	hyp. prot., conserved CDS	0.67	15,365	14.88	6.09	18,147	11.41
	10520	hyp. prot., conserved CDS	0.43	1305	12.19	0.65	2143	11.62
	12830	SAG family member CDS	26.33	2587	7.19	7.27	3773	9.15
	14920	myosin light chain TgMLC1, putative CDS	3.38	1880	9.62	4.21	2252	9.07
	16000	hyp. prot., conserved CDS	0.73	5857	13.40	3.61	8670	11.05
	17610	microneme protein, putative CDS	0.88	1002	10.67	1.53	1053	9.40
	17740	hyp. prot., conserved CDS	1.05	1763	11.21	1.74	2927	10.68
	17850	hyp. prot. CDS	1.63	1067	9.89	3.10	1209	8.67
	23210	hyp. prot. CDS	0.97	11,037	13.93	6.07	12,453	10.82
	24670	hyp. prot., conserved CDS	411.58	2420	3.13	346.08	3561	3.50
	*25290*	*isoleucyl-tRNA synthetase-related, related CDS*	*49,929.39*	*5217*	*−2.69*	*19,238.79*	*4364*	*−1.80*
	25900	hyp. prot., conserved CDS	2.45	1083	9.37	1.73	1213	9.45
	27480	microneme protein etmic-2/7h, related CDS	196.86	1181	3.17	170.74	1716	3.44
	27550	hyp. prot., conserved CDS	19.79	1363	6.68	37.14	1660	5.62
	28320	hyp. prot., conserved CDS	7.57	1958	8.59	7.43	3833	9.11
	29270	hyp. prot., conserved CDS	5.46	1293	8.46	2.71	1779	9.37
	29320	hyp. prot., conserved CDS	1.55	1338	10.26	1.36	1633	10.28
	*29600*	*oocyst wall protein COWP, putative CDS*	*4769.07*	*16,821*	*2.40*	*4277.51*	*15,948*	*2.13*
	30150	hyp. prot., conserved CDS	4.12	1048	8.53	6.06	1610	8.05
	35290	hyp. prot., conserved CDS	3.66	2683	10.02	6.86	5106	9.58
	36540	microneme protein, putative CDS	3.49	1445	9.23	4.58	1243	8.17
	40720	hyp. prot., conserved CDS	722.88	2081	2.10	743.49	2311	1.86
	44080	hyp. prot., conserved CDS	141.85	1028	3.44	144.99	1015	2.94
	*45300*	*MAP3K epsilon protein kinase, related CDS*	*1220.26*	*3758*	*2.20*	*1045.31*	*3708*	*2.05*
	48180	hyp. prot., conserved CDS	2.94	2666	10.34	5.25	3294	9.05
	49350	hyp. prot. CDS	9.83	2652	8.63	17.57	4786	8.21
	*57080*	*hyp. prot., conserved CDS*	*64,395.19*	*6815*	*−2.68*	*26,585.41*	*5728*	*−1.87*

hyp. prot. = hypothetical proteins. Italicized genes (*n* = 4) are shared and significantly differentially expressed between strains with mean TPM > 1000 in a strain at T0 or T36. * Differentially expressed genes (DEGs)—genes in common that were significantly differentially expressed more in either strain (T36 vs. T0) in DESeq pairwise comparisons, meeting thresholds of >1.5 or <−1.5 log_2_ fold change (FC) and adjusted *p* < 0.05 with >1000 overall mean TPM at T0 or T36 in each strain. ** Mean TPM = Transcripts Per Million, averaged among T0 or T36 replicates in each strain.

## Data Availability

The data discussed in this publication have been deposited into NCBI’s Gene Expression Omnibus (GEO) [76] and are accessible through GEO Series accession number GSE247034 (https://www.ncbi.nlm.nih.gov/geo/query/acc.cgi?acc=GSE247034 (accessed on 10 November 2023)).

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
