# Peer review of "RNA-Seq of Phenotypically Distinct Eimeria maxima Strains Reveals Coordinated and Contrasting Maturation and Shared Sporogonic Biomarkers with Eimeria acervulina"

_pathogens, 2023, doi:10.3390/pathogens13010002_

Round 1
Reviewer 1 Report
Comments and Suggestions for Authors
Comments
The authors provide a good knowledge of the current science in the understanding ‘Coordinated and contrasting maturation of phenotypically distinct strains of Eimeria maxima’ I recommend some changes. See below.
Line 23-26: Result needed to clear more preciesly
Line 63-74: Add the reference and write down in a paragraph
Line 253-254: Short the subtitle
Line 334: It’s better (E. acervulina) to describe in details in discussion portion with your results
Line 349-350: Why statistical bars are not mentioned in this diagram
Line 362-365: Explain with reference. I think explain in discussion part rather than in result parts
Line 649-650: Need to explain in discussion part
Line 1072-1073: A. Cyclophilins catalyze protein isomerization and are involved in proper protein folding; they are critical for pathogen infection ‘here” A. Cyclophilins should be italic
1031-1050: Need to explain with a references
Line 952-953: Statistical bars are not cleared in this diagram
Check the references following the journal format
Reviewer 2 Report
Comments and Suggestions for Authors
The manuscript authored by Tucker et al. reported the gene expression profiles of two phenotypically different E. maxima during the sporulation of oocysts. Overall, the manuscript is well-written and readable. The authors firstly, investigated how two strains exhibit differences in pathogenicity regarding the transcript expression profiles during the sporulation, but seems no critical differences were found, then the authors more focus on the differences between before and after the sporulation. The part that the authors compared the result with another Eimeria species, E. acervulina from their previous publication sounds okay, but based on the findings, it is suggested to reshape the title supporting their important findings. Please refer to some minor comments below.
Throughout the manuscript, the authors mentioned other coccidia, such as Cyclospora, several times. It is understandable, but it is suggested to reduce mentioning them as they are different families and have their own maturation process of oocyst.
Please remove reference No. 20 and consider adding the unpublished data supporting what the authors described in the manuscript.
Round 2
Reviewer 1 Report
Comments and Suggestions for Authors
Fine